# Variability and Uncertainty in Flux-Site Scale Net Ecosystem Exchange Simulations Based on Machine Learning and Remote Sensing: A Systematic Evaluation

Haiyang Shi[1,2,4,5], Geping Luo[1,2,3,5], Olaf Hellwich[6], Mingjuan Xie[1,2,4,5], Chen Zhang[1,2], Yu Zhang[1,2], Yuangang Wang[1,2], Xiuliang Yuan[1], Xiaofei Ma[1], Wenqiang Zhang[1,2,4,5], Alishir Kurban[1,2,3,5], Philippe De Maeyer[1,2,4,5] and Tim Van de Voorde[4,5]

[1] State Key Laboratory of Desert and Oasis Ecology, Xinjiang Institute of Ecology and Geography, Chinese Academy of Sciences, Urumqi, Xinjiang, 830011, China.
[2] University of Chinese Academy of Sciences, 19 (A) Yuquan Road, Beijing, 100049, China.
[3] Research Centre for Ecology and Environment of Central Asia, Chinese Academy of Sciences, Urumqi, China.
[4] Department of Geography, Ghent University, Ghent 9000, Belgium.
[5] Sino-Belgian Joint Laboratory of Geo-Information, Ghent, Belgium and Urumqi, China.
[6] Department of Computer Vision & Remote Sensing, Technische Universität Berlin, 10587 Berlin, Germany.

***Correspondence to***: Geping Luo (luogp@ms.xjb.ac.cn) and Olaf Hellwich (olaf.hellwich@tu-berlin.de)

Submitted to *Biogeosciences*

**Abstract.** Net ecosystem exchange (NEE) is an important indicator of carbon cycling in terrestrial ecosystems. Many previous studies have combined flux observations, meteorological, biophysical, and ancillary predictors using machine learning to simulate the site-scale NEE. However, systematic evaluation of the performance of such models is limited. Therefore, we performed a meta-analysis of these NEE simulations. A total of 40 such studies and 178 model records were included. The impacts of various features throughout the modeling process on the accuracy of the model were evaluated. Random Forests and Support Vector Machines performed better than other algorithms. Models with larger time scales have lower average R-squared, especially when the time scale exceeds the monthly scale. Half-hourly models (average R-squared = 0.73) were significantly more accurate than daily models (average R-squared = 0.5). There are significant differences in the predictors used and their impacts on model accuracy for different plant functional types (PFTs). Studies at continental and global scales (average R-squared = 0.37) with multiple PFTs, more sites, and a large span of years correspond to lower R-squared than studies at local (average R-squared = 0.69) and regional scales (average R-squared = 0.7). Also, the site-scale NEE predictions need more focus on the internal heterogeneity of the NEE dataset and the matching of the training set and validation set.

## 1 Introduction

Net ecosystem exchange (NEE) of $CO_2$ is an important indicator of carbon cycling in terrestrial ecosystems (Fu et al., 2019), and accurate estimation of NEE is important for the development of global carbon neutral policies. Although process-based models have been used for NEE simulations (Mitchell et al., 2009), their accuracy and spatial resolutions of the model outputs are limited probably due to the lack of understanding and quantification of complex processes. Many researchers have tried to use a data-driven approach as an alternative (Fu et al., 2014; Tian et al., 2017; Tramontana et al., 2016; Jung et al., 2011). On the one hand, it was made possible by the increase in the growth of global carbon flux observations and the large amount of flux observation data being accumulated. Since the 1990s, the use of the eddy covariance technique to monitor NEE has been rapidly promoted (Baldocchi, 2003). Several regional and global flux measurement networks have been established for the big data management of the flux sites, including CarboEuro-flux (Europe), AmeriFlux (North America), OzFlux (Australia), ChinaFlux (China), FLUXNET (global), etc. On the other hand, machine learning approaches are increasingly used to extract patterns and insights from the ever-increasing stream of geospatial data (Reichstein et al., 2019). The rapid development of various algorithms and high public availability of model tools in the field of machine learning have made these techniques easily available to more researchers in the field of geography and ecology (Reichstein et al., 2019). Since the above two major advances (i.e., increasing availability of flux data and machine learning techniques) in the last two decades, various machine learning algorithms have been used to simulate NEE at the flux station scale with various predictor variables (e.g., meteorological variables, biophysical variables) incorporated for spatial and temporal mapping of NEE or understanding the driving mechanisms of NEE.

To date, studies on using machine learning to predict NEE have a high diversity in terms of modeling approaches. To obtain a comprehensive understanding of machine learning-based NEE prediction, a synthesis evaluation of these machine learning models is necessary. Since the beginning of this century, when machine

learning approaches were still rarely used in geography and ecology research, neural networks were already
used to perform simulations and mapping of NEE in European forests (Papale and Valentini, 2003).
Subsequently, considerable efforts have been made by researchers to improve such predictive models. Many
studies have demonstrated the effectiveness of their proposed improvements (i.e., using predictors with a higher
spatial resolution (Reitz et al., 2021) and using data from the local flux site network (Cho et al., 2021)) by
comparing with previous studies. However, the improvements achieved in these studies may be limited to
smaller areas and specific conditions and may not be generalizable (Cleverly et al., 2020; Reed et al., 2021; Cho
et al., 2021). We are more interested in guidelines with universal applicability that improve the model accuracy,
such as the selection of appropriate predictors and algorithms under different conditions. Therefore, we should
synthesize the results of models applied to different conditions and regions to obtain general insights.

Many factors may affect the performance of these NEE prediction models, such as the predictor variables, the
spatial and temporal span of the observed flux data, the plant functional type (PFT) of the flux sites, the model
validation method, the machine learning algorithm used, as described below:
a)  Predictors: Various biophysical variables (Zeng et al., 2020; Cui et al., 2021; Huemmrich et al., 2019) and
other meteorological and environmental factors have been used in the simulation of NEE. The most
commonly used predictor variables include precipitation (Prec), air temperature (Ta), wind speed (Ws),
net/sun radiation (Rn/Rs), soil temperature (Ts), soil texture, soil moisture (SM) (Zhou et al., 2020), vapor-
pressure deficit (VPD) (Moffat et al., 2010; Park et al., 2018), the fraction of absorbed photosynthetically
active radiation (FAPAR) (Park et al., 2018; Tian et al., 2017), vegetation index (e.g., NDVI, EVI), LAI,
and evapotranspiration (ET) (Berryman et al., 2018). The predictor variables used vary with the natural
conditions and vegetation functional types of the study area. In contrast, in models that include multiple
PFTs, some variables that play a significant role in the prediction of each of the multiple PFTs may have
higher importance. For example, growing degree days (GDD) may be a more effective variable for NEE of
tundra in the northern hemisphere high latitudes (Virkkala et al., 2021), while measured groundwater levels
may be important for wetlands (Zhang et al., 2021). Some of these predictor variables are measured at flux
stations (e.g., meteorological factors such as precipitation and temperature), while others are extracted
from reanalyzed meteorological datasets and satellite remote sensing image data (e.g., vegetation indices).
The spatial and temporal resolution of predictors can lead to differences in their relevance to NEE
observations. Most measured in situ meteorological factors have a good spatio-temporal match to the
observed NEE (site scale, half-hourly scale). However, the proportion of NEE explained by remotely
sensed biophysical covariates may depend on their spatial and time scales. For example, the MODIS-based
8-daily NDVI data may better capture temporal variation in the relationship between NEE and vegetation
growth than the Landsat-based 16-daily NDVI data. In contrast, the interpretation of NEE by variables
such as soil texture and soil organic content (SOC), which do not have temporal dynamic information, may
be limited to the interpretation of spatial variability, although they are considered to be important drivers of
NEE. Therefore, the importance of variables obtained from NEE simulations based on a data-driven
approach may differ from that in process-based models as well as in the actual driving mechanisms. This
may be related to the spatial and temporal resolution of the predictors used and the quality of the data. It is
necessary to consider the spatio-temporal resolution of the data for the actual biophysical variables used in
the different studies in the systematic evaluation of data-driven NEE simulations.
b)  The spatio-temporal heterogeneity of data sets, and validation method: The spatio-temporal heterogeneity
of the dataset may affect model accuracy. Typically, training data with larger regions, multiple sites,
multiple PFTs, and longer spans of years may have a higher degree of imbalance (Kaur et al., 2019; Van
Hulse et al., 2007; Virkkala et al., 2021; Zeng et al., 2020). Modeling with unbalanced data (where the
difference between the distribution of the training and validation sets is significant even if selected at
random) may result in lower model accuracy. To date, the most commonly used methods for validating
such models include spatial (Virkkala et al., 2021), temporal (Reed et al., 2021), and random (Cui et al.,
2021) cross-validation. The imbalance of data between the training and validation sets may affect the
accuracy of the models when using these validation methods. Spatial validation is used to assess the ability
of the model to adapt to different regions or flux sites of different PFTs, and a common method is 'leave
one site out' cross-validation (Virkkala et al., 2021; Zeng et al., 2020). If the data from the site left out is
not covered (or partially covered) by the distribution of the training dataset, the model's prediction
performance at that site may be poor due to the absence of a similar type in the training set. Temporal
validation typically uses some years of data as training and the remaining years as validation to assess the
model's fitness for interannual variability. For a year that is left out (e.g. a special extreme drought year
which does not occur in the training set), the accuracy of the model may be limited if there are no similar
years (extreme drought years) in the training dataset. K-fold cross-validation is commonly used in random
cross-validation to assess the fitness of the model to the spatio-temporal variability. In this case, different
values of K may also have a significant impact on the model accuracy. For example, for an unbalanced
dataset, the average model accuracy obtained from a 10-fold (K = 10) validation approach is likely to be
higher than that of a 3-fold  (K = 3) validation approach (Marcot and Hanea, 2021).
c)  Machine learning algorithms used: Simulating NEE using different machine learning algorithms may
influence the model accuracy, which may be induced by the characteristics of these algorithms themselves
and the specific data distribution of the NEE training set. For example, Neural Networks can be used
effectively to deal with nonlinearities, while as an ensemble learning method, Random Forests can avoid
overfitting due to the introduction of randomness. Therefore, a comprehensive evaluation of this is
necessary.

In this study, to evaluate the impacts of predictors use, algorithms, spatial/time scale, and validation methods on
model accuracy, we performed a meta-analysis of papers with prediction models that combine NEE
observations from flux towers, various predictors, and machine learning for the data-driven NEE simulations. In
addition, we also analyzed the causality of multiple features in NEE simulations and the joint effects of multiple
features on model accuracy using the Bayesian Network (BN) (a multivariate statistical analysis approach
(Pearl, 1985)). The findings of this study can provide some general guidance for future NEE simulations.
**2 Methodology**
**2.1 Criteria for including articles**
In the Scopus database, a literature query was applied to titles, abstracts, and keywords (Table 1) according to
Preferred Reporting Items for Systematic Reviews and Meta-Analyses (PRISMA) (Moher et al., 2009) (Fig. 1):
a) Articles were filtered for those that modeled NEE. Articles that modeled other carbon fluxes such as
methane flux were not included.
b) Articles that used only univariate regression rather than multiple regression were screened out.
c) Articles reported the determination coefficient (R-squared) of the validation step (Shi et al., 2021;
Tramontana et al., 2016; Zeng et al., 2020) as the measure of model performance. Although RMSE is also
often used for model accuracy assessment, its dependence on the magnitude of water flux values makes it
difficult to use for fair comparisons between studies.
d) Articles were published in journals with language limited to English.
e) Articles were filtered for those that were published in the specific journals (Table S1) for research quality
control because the data, model implements, and peer review in these journals are often more reliable.

Table 1. Article search query design: '[A1 OR A2 OR A3...] AND [B1 OR B2...] AND [C1 OR C2...]'

| ID | A | B | C |
|----|---|---|---|
| 1 | Carbon flux | "Eddy covariance" | "machine learning" |
| 2 | $CO_2$ flux | "Flux tower" | regress* |
| 3 | "net ecosystem exchange" | | "Support Vector" |
| 4 | net ecosystem produc | | "Neural Network" |
| 5 | gross primary produc | | "Random Forest" |
| 6 | Carbon exchange | | |


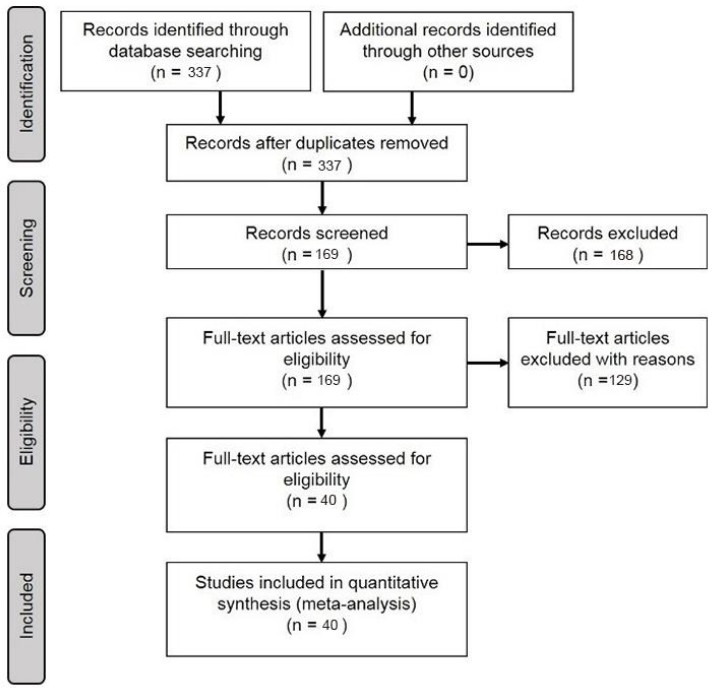


Figure 1. PRISMA-based paper filtering flowchart.

**2.2 Features of prediction models**

Typically, the flow of the NEE prediction modeling framework (Fig. 2) based on flux observations and machine
learning is as follows: first, half-hourly scale NEE flux observations are aggregated into various time scale NEE
data, and gap-filling techniques (Moffat et al., 2007) are often used in this step to obtain complete NEE series
when data are missing. Various predictors including meteorological variables, remote sensing-based biophysical
variables, etc. are extracted to match site-scale NEE series to generate a training dataset containing the target
variable NEE and various covariates. Subsequently, various algorithms are used for the NEE prediction model
construction and validated in different ways (e.g., leave-one-site-out validation (Zeng et al., 2020)). Finally, in
some studies, prediction models were applied to gridded covariate data to map the regional or global-scale NEE
spatial and temporal variations (Zeng et al., 2020; Papale and Valentini, 2003; Jung et al., 2020). The
information of R-squared (at the validation phase) and the associated model features reported in the article are
considered as one data record for the formal meta-analysis (i.e., each R-squared record corresponding to a
prediction model). From the included papers, R-squared records and various features (Table 2) involved in the
NEE modeling framework (Fig. 2) were extracted (including the used algorithms, modeling/validation methods,
remote sensing data, meteorological data, biophysical data, and ancillary data). In some studies, multiple
algorithms were applied to the same dataset, or models with different features were developed (Virkkala et al.,
2021; Zhang et al., 2021; Cleverly et al., 2020; Tramontana et al., 2016). In these cases, multiple data records
will be documented.

In the practical information extracting step, we categorized such features in a comparable manner. First, we
categorized the various algorithms used in these papers, although the same algorithm may also have a variant
form or an optimized parameter scheme. They are categorized into the following families of algorithms:
Random Forests (RF), Multiple Linear Regressions (MLR), Artificial Neural Networks (ANN), Support Vector
Machines (SVM), Partial Least Squares Regression (PLSR), Generalized additive model (GAM), Boosted
Regression Tree (BRT), Bayesian Additive Regression Trees (BART), Cubist, model tree ensembles (MTE).
Second, we classified the spatial scales of these studies. Models with study areas (spatial extent covered by flux
stations) smaller than 100x100 km were classified as 'local' scale models, those with study area sizes exceeding
continental scale were classified as 'global' scale, and those with study area sizes in between were classified as
'regional' scale. Third, for various predictors, we only recorded whether the predictors were used or not without
distinguishing the detailed data sources and categories (e.g., grid meteorological data from various reanalysis
datasets and in-situ meteorological observations from flux stations), measurement methods (e.g., soil moisture
measured/estimated by remote sensing or in situ sensors), etc. Fourth, we documented PFTs for the prediction
models from the description of study areas or sites in these papers. They are classified into the following types:
forest, grassland, cropland, wetland, savannah, tundra, and multi-PFTs (models containing a mixture of multiple
PFTs). Models not belonging to the above PFTs were not given a PFT field and were not included in the
subsequent analysis of the PFT differences. Other features (Table 2) are extracted directly from the
corresponding descriptions in the papers in an explicit manner.

Subsequently, the model accuracies corresponding to different levels of various features are compared in a
cross-study fashion. In the evaluation of algorithms and time scales, we also implement comparisons within
individual studies. For example, in the evaluation of the effects of the algorithms, we compare the accuracy of
models using the same training data and keeping other features as constants in individual studies. In this intra-
study comparison step, only algorithms with relatively large sample sizes in the cross-study comparisons were
selected. In this study, algorithms with less than 10 available model records are not considered to have a
sufficient sample size and we do not give further conclusive opinions on the accuracy of these algorithms due to
their small samples (e.g., PLSR and BART with high R-squared but very few records as evidence). MLR, RF,
SVM, and ANN were found to have large sample sizes (Fig. 5a), and thus their accuracies can be comparable.
Based on this, in the intra-study comparison step, we only compare the accuracy differences between MLR, RF,
SVM, and ANN in the context of using the same data and the same other model features (Fig. 5b).

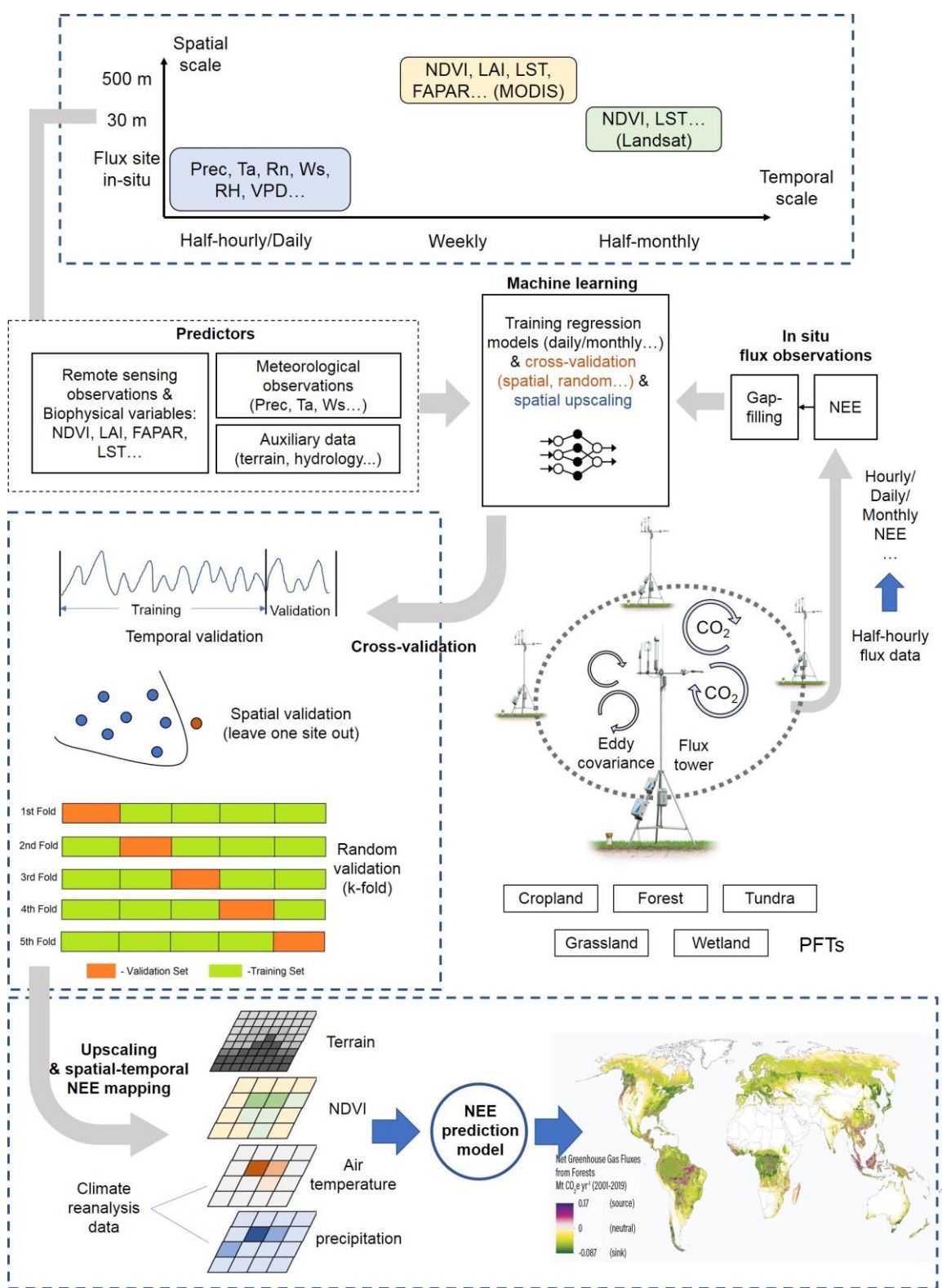


Figure 2. Features of the machine learning-based NEE prediction process. The flux tower photo is from

https://www.licor.com/env/support/Eddy-Covariance/videos/ec-method-02.html (last accessed: 23rd March

2022). The map in the lower part is from Harris et al., 2021. Prec, Ta, Rn, Ws, RH, and VPD represent

precipitation, air temperature, net surface radiation, wind speed, relative humidity, and vapour-pressure deficit

respectively. FAPAR is the fraction of absorbed photosynthetically active radiation. LST is the land surface

temperature. LAI is the leaf area index.

207

208    Table 2. Description of information extracted from the included papers.

| Field/Feature | Definition | Categories adopted |
| --- | --- | --- |
| Id paper | Identification number of the paper (internal) | |
| Paper | Paper metadata | |
| Author/s | Name/s of author/s | |
| Title | Title of the paper | |
| Year | Year of publication | |
| Publication title | Name of the journal where the paper was published | |
| Plant functional type (PFT) | PFTs for the flux sites used | 1-forest, 2-grassland, 3-cropland, 4-wetland, 5-savannah, 6-tundra and multi-PFTs |
| Location | More precise location (with the latitude and longitude of the center of the studied sites). Global (mainly based on FluxNet (Tramontana et al., 2016)) and continental-scale studies are not shown on the map due to the difficulty of identifying specific locations. | latitude, longitude |
| Algorithms | Algorithm families used in the multivariate regression | Random Forests (RF), Multiple Linear Regressions (MLR), Artificial Neural Networks (ANN), Support Vector Machines (SVM), Partial Least Squares Regression (PLSR), Generalized additive model (GAM), Boosted Regression Tree (BRT), Bayesian Additive Regression Trees (BART), Cubist, model tree ensembles (MTE). |
| Sites number | Number of the flux sites used | |
| Study area/Spatial scale | Area representatively covered by the flux sites | local (less than $100 \times 100$ km), regional, global (continent-scale and global scale) |
| Time scale | The time scale of the model | half-hourly, hourly, daily, weekly, 8-daily, monthly, seasonally, yearly |
| Study period | The period of the data used in the model | year, growing season, daytime, spring, summer, autumn, winter |
| Year span | The span of years of the flux data used | |
| Site year | Describe the volume of total flux data with the number of sites and years aggregated. | |
| Cross-validation | Describe the chosen method of cross-validation. | Spatial (e.g., 'leave one site out'), temporal (e.g., 'leave one year out'), random (e.g., 'k-fold') |

| | | |
|---|---|---|
| Training/validation | Describe the ratio of the data in training and validation sets. | |
| Satellite images | Describe the source of satellite images used to derive NDVI, EVI, LAI, LST, etc. | Landsat, MODIS, Hyperion (EO-1), AVHRR, IKONOS |
| Biophysical predictors | LAI, NDVI/EVI, evapotranspiration (ET) (i.e., the latent heat observed by the flux station), enhanced vegetation index (EVI), the fraction of absorbed photosynthetically active radiation/photosynthetically active radiation (FAPAR/PAR), leaf area index (LAI), etc. | Used (recorded as '1') or not used (recorded as '0') |
| Meteorological variables | precipitation (Prec), net radiation/solar radiation (Rn/Rs), air temperature (Ta), vapour-pressure deficit (VPD), relative humidity (RH) , etc. | Used (recorded as '1') or not used (recorded as '0') |
| Ancillary data | Describe the source of ancillary variables including terrain variables derived from DEM, soil texture, or hydrology-related data: soil organic content (SOC), soil texture, terrain, soil moisture/land surface water index (SM_LSWI), etc. | Used (recorded as '1') or not used (recorded as '0') |
| Top three variables in the ranking of importance of predictors | Describe the interpretation of the importance of variables in machine learning models. | |
| Accuracy measure | Accuracy measure used to assess the performance of the estimation/prediction | R-squared (in the validation phase) |

209

## 2.3 Bayesian Network for analyzing joint effects

Based on the Bayesian network (BN), the joint impacts of multiple model features on the R-squared are
analyzed. A BN can be represented by nodes $(X_1,., X_n)$ and the joint distribution (Pearl, 1985):

$$P(X) = P(X_1, X_2, \dots, X_n) = \prod_{i=1}^{n} P(X_i | pa(X_i)) \qquad (1)$$

where $pa(X_i)$ is the probability of the parent node $X_i$. Expectation-maximization (EM) approach (Moon, 1996) is
used to incorporate the collected model records and compile the BN.

Sensitivity analysis is used for the evaluation of node influence based on mutual information (MI) which is
calculated as the entropy reduction of the child node resulting from changes at the parent node (Shi et al., 2020):

$$\text{MI} = \text{H(Q)-H(Q|F)} = \sum_q \sum_f P(q, f) \log_2 \left( \frac{P(q,f)}{P(q)P(f)} \right) \qquad (2)$$

where H represents the entropy, Q represents the target node, F represents the set of other nodes and q and f

represent the status of Q and F.

## 3 Results

### 3.1 Articles included in the meta-analysis

We included 40 articles (Table S2) and extracted 178 model records for the formal meta-analysis (Fig. 1). Most

studies were implemented in Europe, North America, Oceania, and China (Fig. 3). The number of such papers is

increasing recently (Fig. 4) and it shows the machine learning approach for NEE prediction has been of interest

to more researchers. The main journals in which these articles have been published (Fig. 4) include Remote

Sensing of Environment, Global Change Biology, Agricultural and Forest Meteorology, Biogeosciences, and

Journal of Geophysical Research: Biogeosciences, etc.

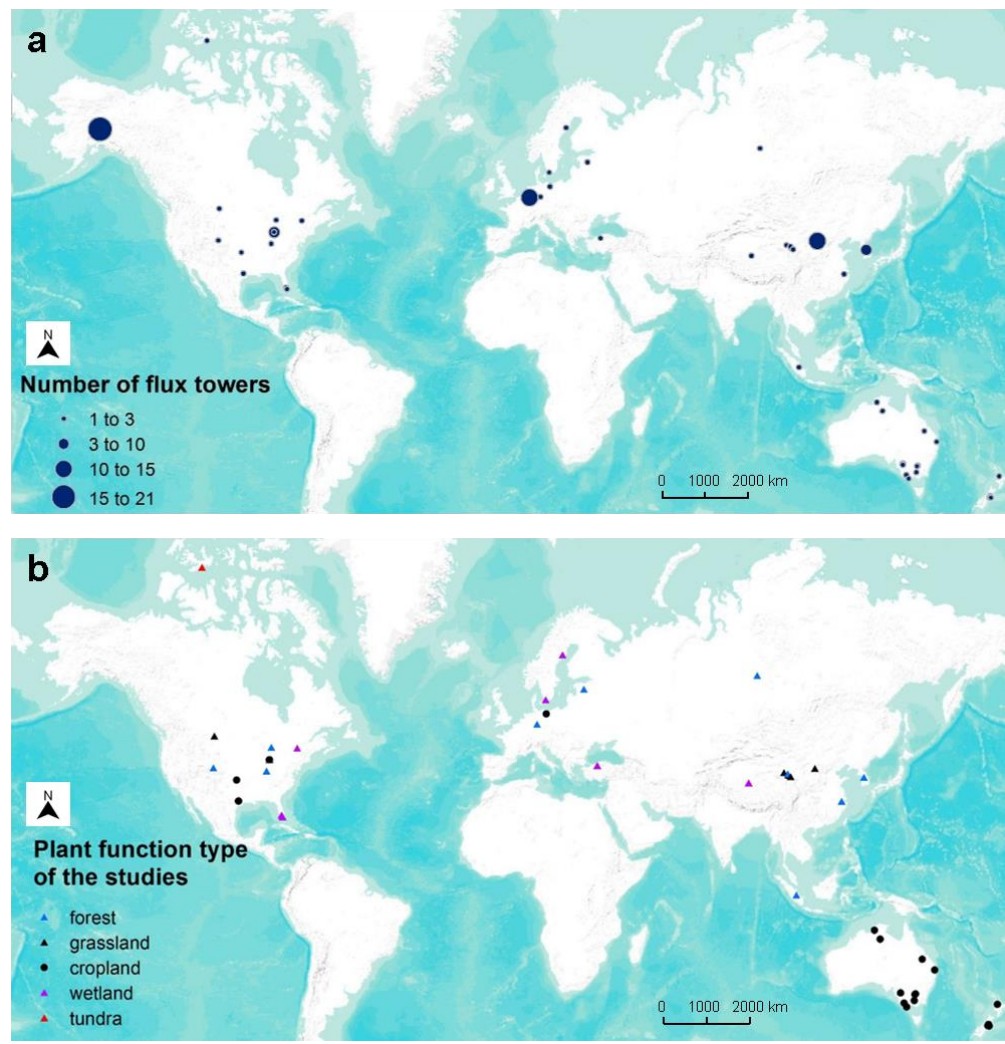

Figure 3. Location of studies (a) included with the number of flux sites included and (b) their PFTs in the meta-

analysis (total of 40 studies and 178 model records). Global (mainly based on FluxNet (Tramontana et al.,

2016)) and continental-scale studies are not shown on the map due to the difficulty of identifying specific
locations.

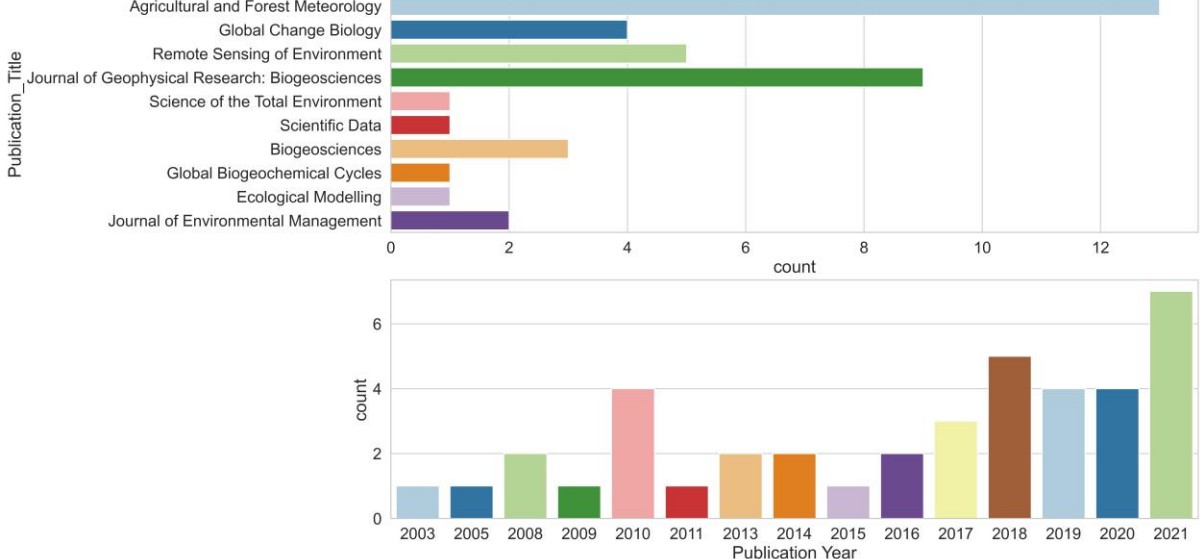


Figure 4. The number of studies published across journals and the total number of publications per year.
**3.2 The formal Meta-analysis**
We assessed the impact of the features (e.g., algorithms, study area, PFTs, amount of data, validation methods,
predictor variables, etc.) used in the different models based on differences in R-squared.
**3.2.1 Algorithms**
Among the more frequently used algorithms, ANN and SVM performed better (Fig. 5a) on average across
studies (lightly better than RF). On the other hand, since cross-study comparisons of algorithm accuracy include
differences in data used in model construction, we performed a pairwise comparison (Fig. 5b) of these four
algorithms (i.e., ANN, SVM, RF, and MLR). In these studies, multiple models are developed for consistent
training data with the interference of training data differences removed. It shows that RF and SVM perform best
in the inter-study comparison (Fig. 5b). Whereas ANN performed slightly worse than RF and SVM, all three of
them were stronger than MLR. Overall, the performance of RF and SVM may be good and similar in the NEE
simulations.

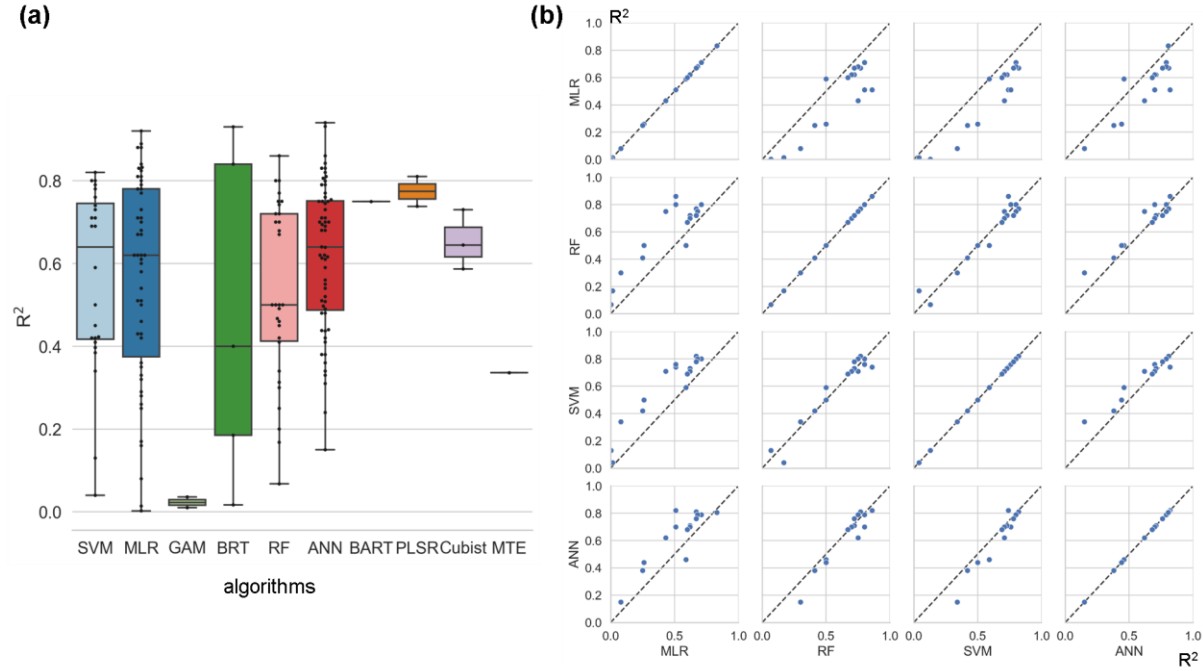

Figure 5. Differences in model accuracy (R-squared) using different algorithms across studies (a) and internal comparisons of the model accuracy (R-squared) of selected pairs of algorithms within individual studies (b). Regression algorithms: Random Forests (RF), Multiple Linear Regressions (MLR), Artificial Neural Networks (ANN), Support Vector Machines (SVM), Partial Least Squares Regression (PLSR), Generalized additive model (GAM), Boosted Regression Tree (BRT), Bayesian Additive Regression Trees (BART), Cubist, model tree ensembles (MTE). In panel (a), the horizontal line in the box indicates the medians. The top and bottom border lines of the box indicate the 75% and 25% percentiles, respectively.

### 3.2.2 Time scales

The impact of time scale on R-squared is considerable (Fig. 6), with models with larger time scales having lower average R-squared, especially when the time scale exceeds the monthly scale. The most frequently used scales were the daily, 8-day, and monthly scales. In studies where multiple time scales were used with other characteristics being the same, we found that models with half-hourly scales were significantly more accurate than models with daily scales (Fig. 6). However, the difference in accuracy between the day-scale and week-scale models is small. The accuracy of models with a monthly scale is the lowest.

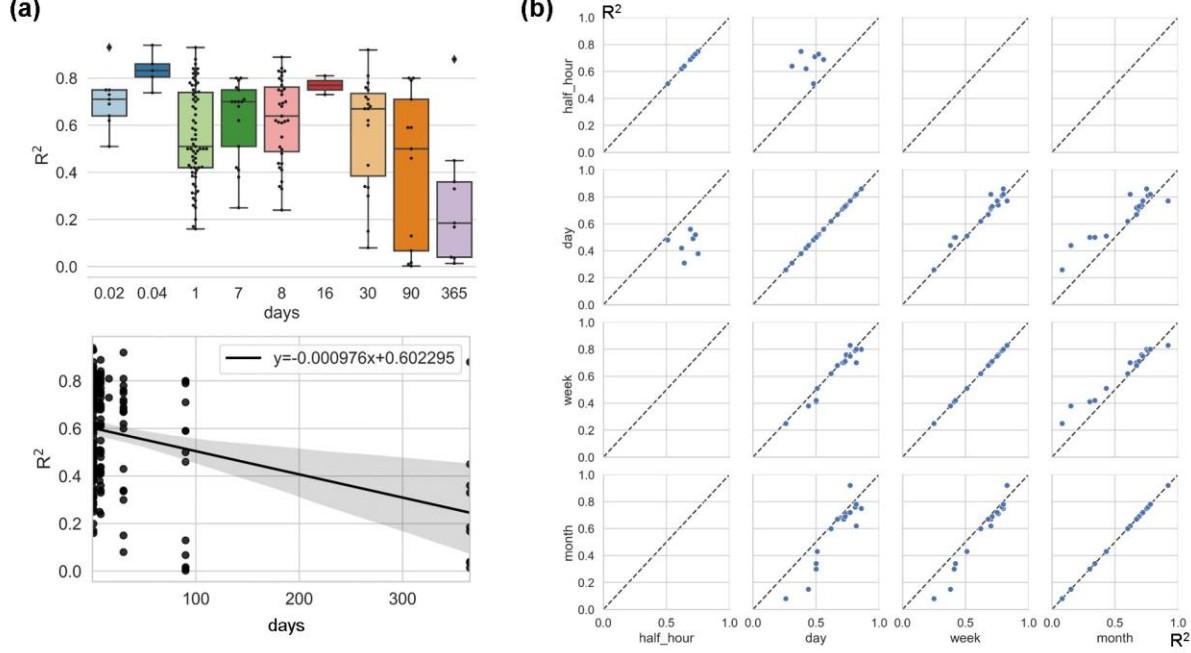


Figure 6. Differences in model accuracy (R-squared) at different time scales across studies with the
linear regression between R-squared and time scales (a), and comparison of the model accuracy (R-
squared) of selected pairs of time scales within individual studies (b). All model records were
included in panel (a), while studies that used multiple time scales (with other model characteristics
unchanged) were included in panel (b). Time scales: 0.02 days (half-hourly), 0.04 days (hourly), 30
days (monthly), and 90 days (quarterly).
**3.2.3 Various predictors**
Among the commonly used predictors for NEE, there are significant differences in the predictors used and their
impacts on model accuracy for different PFTs (Fig. 7). Ancillary data (e.g. soil texture, soil organic content,
topography) that do not have temporal variability are used less frequently because they can only explain spatial
heterogeneity. In contrast, the biophysical variables LAI, FAPAR, and ET were used significantly less
frequently than NDVI/EVI, especially in the cropland and wetland types. The meteorological variables Ta,
Rn/Rs, and VPD were used most frequently. For forest sites, Rn/Rs and Ws appear to be the variables that
improve model accuracy. For grassland sites, we found that NDVI/EVI appears to be the most effective, despite
the small sample size. For sites in croplands and wetlands, we did not find predictor variables that had a
significant impact on model accuracy.

For different PFTs, the top three variables in the ranking of model importance differed (Fig. S1). SM, Rn/Rs,
Ta, Ts, and VPD all showed high importance across PFTs. This suggests that the variability of measured site-
scale moisture and temperature conditions is important for the simulation of NEE for all PFTs. In contrast, in the
importance ranking, other variables such as precipitation and NDVI/EVI may not lead because of the lag in their
effect on NEE (Hao et al., 2010; Cranko Page et al., 2022). And some other variables may improve model
accuracy for specific PFTs such as groundwater table depth (GWT) for wetland sites and growing degree days
(GDD) for tundra sites.

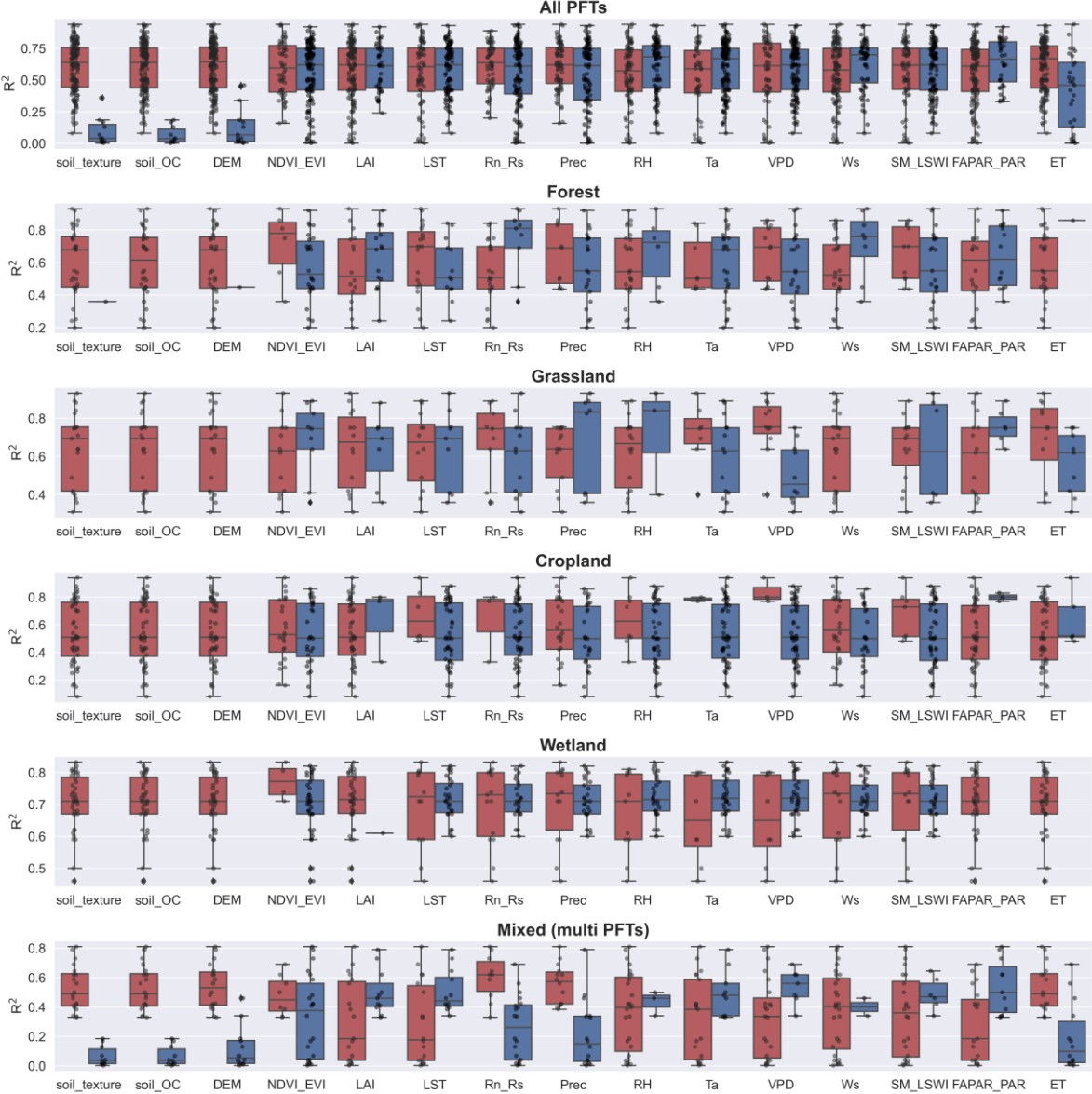

Figure 7. The impact of the various predictors incorporated in models of different PFTs (1-forest, 2-grassland, 3-cropland, 4-wetland, 6-tundra) on R-squared. Dark blue boxes indicate that the predictor was used in the model, while dark red boxes indicate that the predictor was not used. Predictors: soil organic content (Soil_OC), precipitation (Prec), soil moisture/land surface water index (SM_LSWI), net radiation/solar radiation (Rn_Rs), enhanced vegetation index (EVI), air temperature (Ta), vapor-pressure deficit (VPD), the fraction of absorbed photosynthetically active radiation/photosynthetically active radiation (FAPAR_PAR), relative humidity (RH), evapotranspiration (ET), leaf area index (LAI).

### 3.2.4 Other features

In addition, we evaluated other features of the model construction that may contribute to differences in model accuracy (Fig. 8). Studies at continental and global scales with a large number of sites and a large span of years correspond to lower R-squared than studies at local and regional scales, suggesting that studies with a large number of sites across large regions are likely to have high variability in the relationship between NEE and

covariates and that studies at small scales are more likely to have higher model accuracy. Spatial validation
(usually 'leave one site out') corresponds to lower model accuracy compared to random and temporal validation.
This again confirms the dominant role of heterogeneity in the relationship between NEE and covariates across
sites in explaining model accuracy. This seems to be indirectly supported by the fact that a high ratio of training
to validation sets corresponds to a low R-squared, as this high ratio tends to be accompanied by the use of the
'leave one site out' validation approach. The accuracy of the models with a growing season period was slightly
higher than that of the models with an annual period. For the satellite remote sensing data used, the models
based on MODIS data with biophysical variables extracted were slightly less accurate than those based on
Landsat data. For the daily scale models, Landsat data performed a little better than MODIS (Fig. S2). This
suggests that the higher temporal resolution of MODIS compared to Landsat may not play a dominant role in
improving model accuracy. This may also be partially attributed to studies using MODIS-based explanatory data
that tend to include too large surrounding areas around the site (e.g., 2x2 km), which can lead to a scale
mismatch between the flux footprint and the explanatory variables.

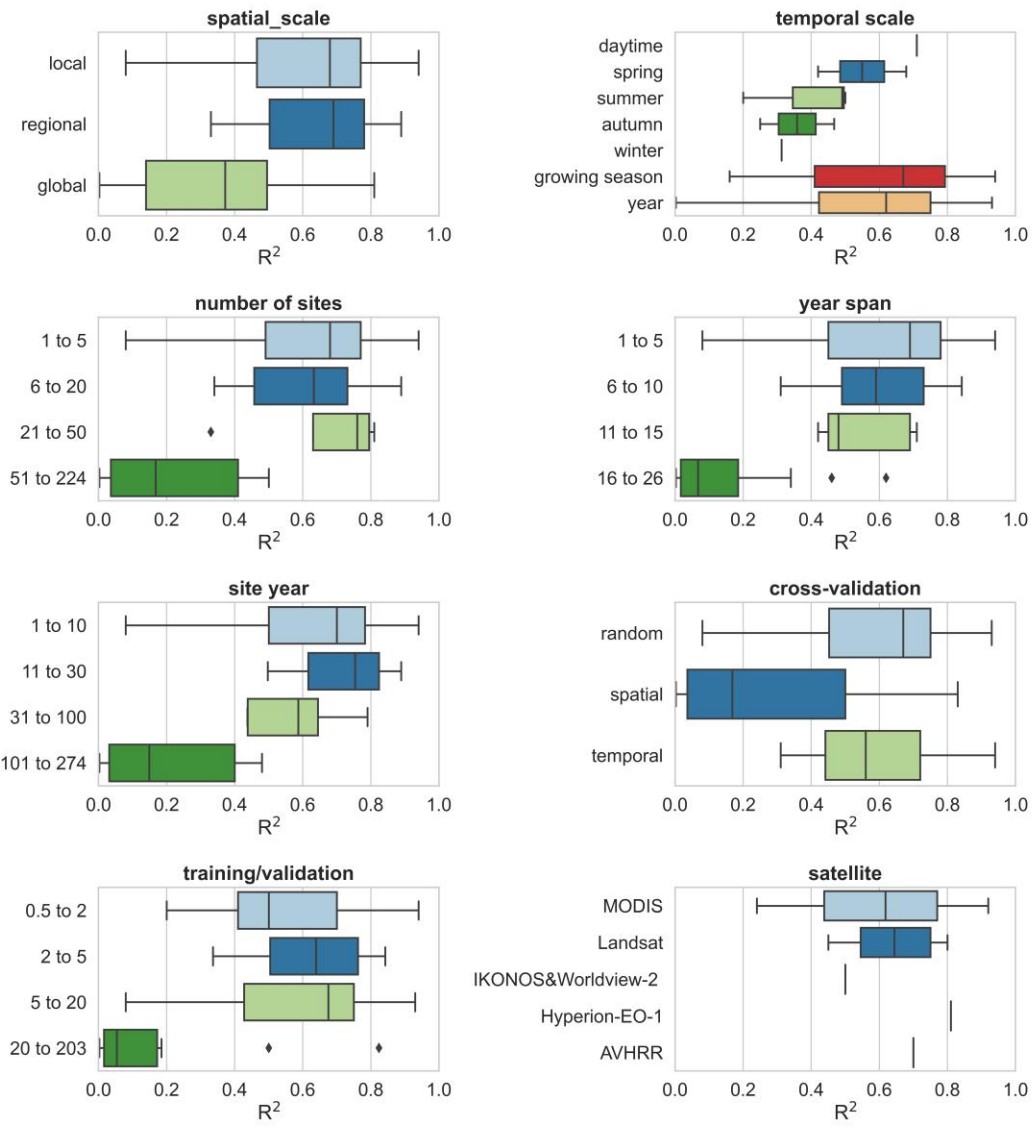


Figure 8. The impacts of other features (i.e. spatial scale, study period, number of sites, year span, site year,
cross-validation method, training/validation, and satellite imagery) on the model performance.

**3.3. The joint causal impacts of multi-features based on the BN**

We selected the features that had a more significant impact on model accuracy in the above assessment and further incorporated them into the BN-based multivariate assessment to understand the joint impact of multiple features on R-squared. The features incorporated included the spatial scale, the number of sites, the time scale, the span of years, the cross-validation method, and whether some specific predictors were used. We discretized the distribution of individual nodes and compiled the BN (Fig. 9a) using records from different PFTs as input. Sensitivity analysis of the R-squared node (Fig. 10) showed that R-squared was most sensitive to 'year span', cross-validation method, Rn/Rs, and time scale under multi-feature control. In the forest and cropland types, R-squared is more sensitive to Rn/Rs, while in the wetland type it is more sensitive to SM/LSWI and Ta. The sensitivity of R-squared to 'year span' was much higher in the cropland type compared to the other PFTs, which may suggest that the interannual variability in the NEE simulations of the cropland type is higher due to potential interannual variability of the planting structure and irrigation practices. For the cropland type, differences in the phenology, harvesting, and irrigation (water volume and frequency) in different years can lead to significant inter-annual differences in NEE simulations. Subsequently, using the constructed BN (with the empirical information in previous studies incorporated), for new studies we can instructively infer the probability distribution of the possible R-squared (Fig. 9b) with some model features predetermined. In previous studies, spatio-temporal mapping of NEE based on statistical models has often lacked accuracy assessment since there are no grid-scale NEE observations, and this BN may have the potential to be used to validate the accuracy (R-squared) of the NEE time series output of the grid-scale (i.e. inferring possible R-squared from model features, where the output of the grid-scale is considered to be of the form 'leave one site out').

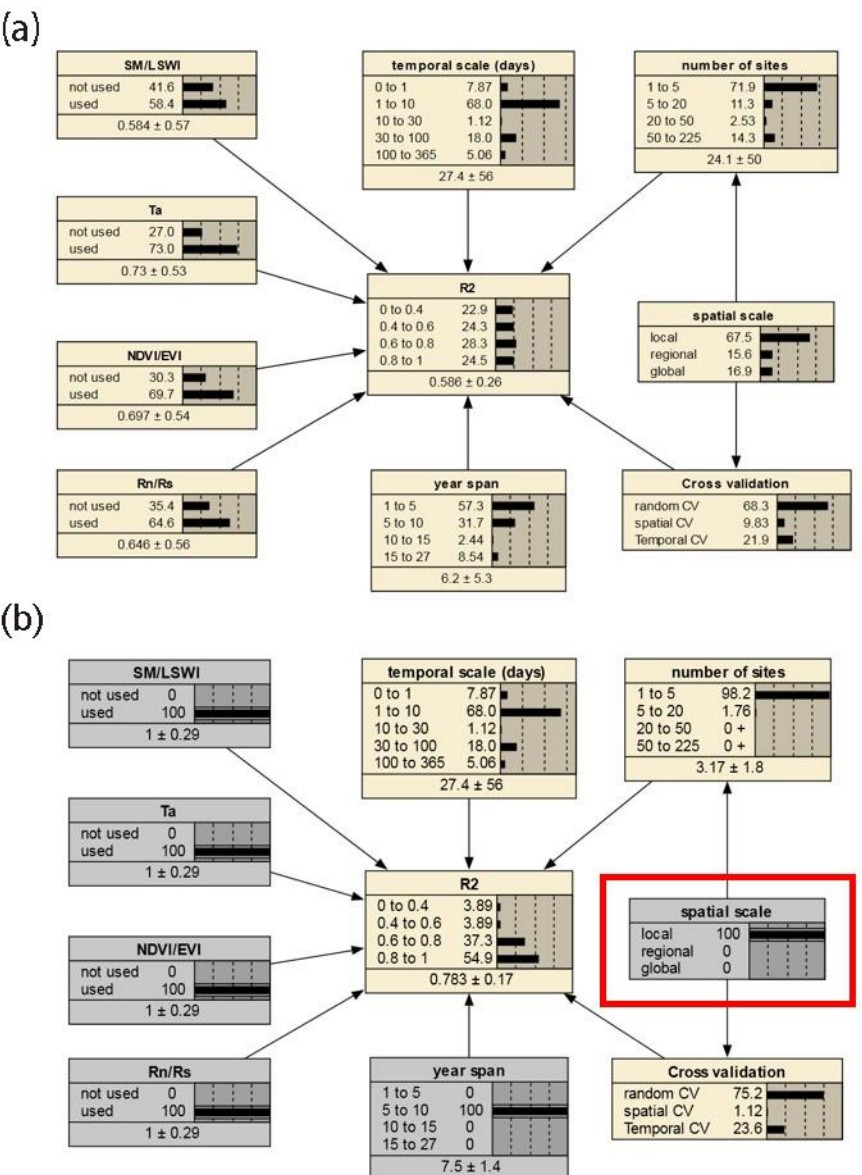


Figure 9. The joint effects of multiple features on the R-squared based on the BN with all records input (a) and
the inference on the probability distribution of R-squared based on the BN with the status of some nodes
determined (b). The values before and after the "±" indicate the mean and standard deviation of the distribution,
respectively. The gray boxes indicate that the status of the nodes has been determined. In panel (b), specific
values of parent nodes such as 'spatial scale' are determined (shown in the red box), leading to an increase in the
expected R-squared compared to the average scenario of panel (a) (as inferred from the posterior conditional
probabilities with the status of the node 'spatial scale' are determined as 'local').

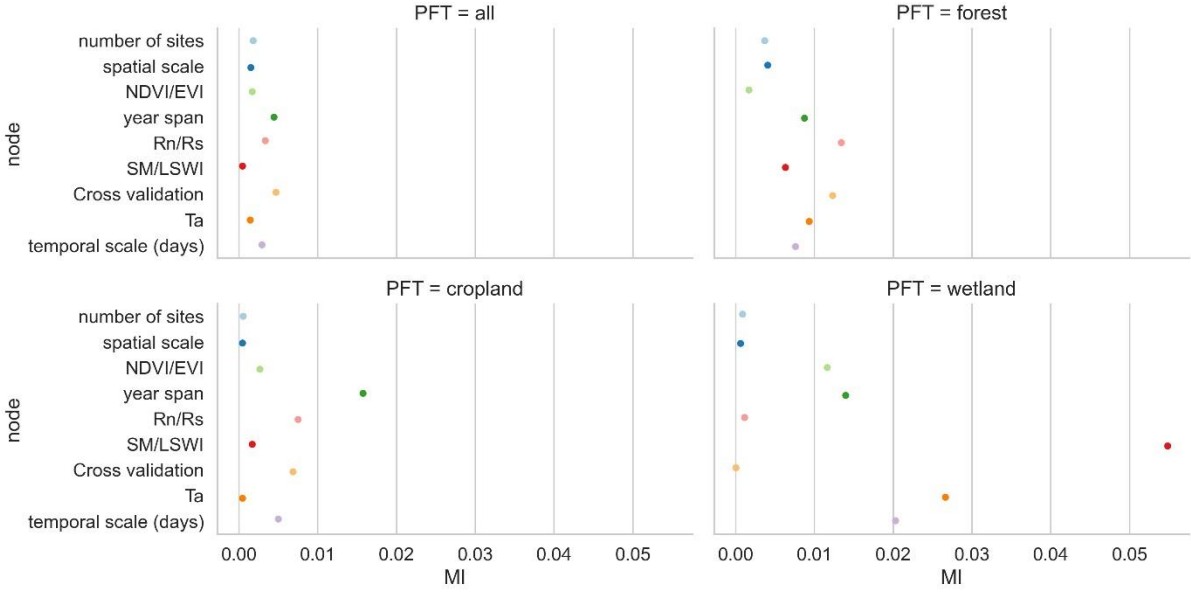


Figure 10. The sensitivity analysis of the R-squared node to other nodes based on the mutual information (MI)
across PFTs. 'Cross-validation' is the cross-validation method including spatial, temporal, and random cross-
validation.

## 4 Discussions

Many studies have evaluated the incorporation of various predictors and model features using machine learning
for improving the site-scale NEE predictions (Tramontana et al., 2016; Zeng et al., 2020; Jung et al., 2011). A
comprehensive evaluation of these studies to provide definitive guidance on the selection of features in NEE
prediction modeling is limited. This study fills the research gap with a meta-analysis of the literature through
statistics on the accuracy and performance of models. Machine learning-based NEE simulations and predictions
still suffer from high uncertainty. By better understanding the expected improvements that can be achieved
through the inclusion of different features, we can identify priorities for the consideration of different features in
modeling efforts and avoid operations decreasing model accuracy.

Compared to previous comparisons of machine learning-based NEE prediction models, this study is more
comprehensive. Previous studies (Abbasian et al., 2022) have also found advantages of RF over other
algorithms in NEE prediction. This study consolidated this finding using a larger amount of evidence. Previous
studies (Tramontana et al., 2016) have also compared the impact of different practices in NEE prediction models
based on the R-squared, such as comparing the difference in accuracy between the two predictor combinations
(i.e., using only remotely sensed data and using remotely sensed data and meteorological data together). In
contrast, since this study incorporated more detailed factors influencing model accuracy, the understanding of
such issues was deepened. However, there are still many uncertainties and challenges in NEE prediction not
clarified in this study.

### 4.1 Challenges in the site-scale NEE simulation and implications for other carbon flux simulations

#### 4.1.1 Variations in time scales

In the above analysis, we found that the effect of the time scale of the model is considerable. This suggests that we should be careful in determining the time scale of the model to consider whether the predictor variables used will work at this time scale. Previous studies have reported the dependence of the NEE variability and mechanism on the time scales. On the one hand, the importance of variables affecting NEE varies at different time scales. For example, in tropical and subtropical forests in southern China (Yan et al., 2013), seasonal NEE variability is predominantly controlled by soil temperature and moisture, while interannual NEE variability is controlled by the annual precipitation variation. A study (Jung et al., 2017) showed that for annual-scale NEE variability, water availability and temperature were the dominant drivers at the local and global scales, respectively. This indicates the need to recognize the temporal and spatial driving mechanisms of NEE in advance in the development of NEE prediction models. On the other hand, dependence may exist between NEE anomalies at various time scales. For example, previous studies (Luyssaert et al., 2007) showed that short-term temperature anomalies may interpret both the daily and seasonal NEE anomalies. This implies that the models at different time scales may not be independent. In the previous studies, the relationship between prediction models at different scales has not been well investigated, and it may be valuable to compare the relations between data and models at different scales in depth. Larger time scales correspond to lower model accuracy, possibly related to the fact that some small-time-scale relations between NEE and covariates (especially meteorological variables) are smoothed. In particular, for models with time scales smaller than one day (e.g. half-hourly models), the 8-daily and 16-daily biophysical variable data obtained from satellite remote sensing are difficult to explain the temporal variation in the sub-daily NEE. Therefore, for models at small time scales (i.e. half-hourly, hourly, daily scale models), in situ meteorological variables may be more important. The inclusion of some ancillary variables (e.g. soil texture, topographic variables) with no temporal dynamic information may be ineffective unless many sites are included in the model and the spatial variability of the ancillary variables for these sites is sufficiently large (Virkkala et al., 2021).

In terms of completeness and purity of training data, hourly and daily models can be better compared to monthly and yearly models. Hourly and daily models can usually preclude those low-quality data and gaps in the flux observations. However, for monthly and yearly scale models, gap-filling (Ruppert et al., 2006; Moffat et al., 2007; Zhu et al., 2022) is necessary because there are few complete and continuous fluxes observations without data gaps on the monthly to yearly scales. Since various gap-filling techniques rely on environmental factors (Moffat et al., 2007) such as meteorological observations, this may introduce uncertainty in the predictive models (i.e., a small fraction of the observed information of NEE is estimated from a combination of independent variables). How it would affect the accuracy of prediction models at various time scales remains uncertain, although various gap-filling techniques have been widely used in the pre-processing of training data.

In addition, the impacts of lagged effects (Hao et al., 2010; Cranko Page et al., 2022) of covariates are not considered in most models, which may underestimate the degree of explanation of NEE for some predictor variables (e.g. precipitation). Most of the machine learning-based models use only the average Ta and do not take into account the maximum temperature, minimum temperature, daily difference in temperature, etc., as in

the process-based ecological models (Mitchell et al., 2009). This suggests that the inclusion of different
temporal characteristics of individual variables in machine learning-based NEE prediction models may be
insufficient.
**4.1.2 Scale mismatch of explanatory predictors and flux footprints**
An excessively large extraction area of remote sensing data (e.g., 2x2 km) may be inappropriate. In the non-
homogeneous underlying conditions, the agreement of the area of flux footprints with the scale of the predictors
should be considered in the extraction of the predictor variables in various PFTs (Chu et al., 2021).
The effects of this mismatch between explanatory variables and flux footprints may be diverse for different
PFTs. For example, for cropland types, the NEE is monitored at a range of several hundred meters around the
flux towers, but remote sensing variables such as FAPAR, NDVI, LAI, etc. can be extracted at coarse scales
(e.g., 2x2 km), some effects outside the extent of the flux footprint (Chu et al., 2021; Walther et al., 2021) are
incorporated (e.g., planting structures with high spatial heterogeneity, agricultural practices such as irrigation).
And for more homogeneous types such as grasslands, coarse-scale meteorological data may still cause spatial
mismatches, even though the differences in land cover types within the 2x2 km and 200x200 m extent around
the flux stations in grasslands may not be considerable. For example, precipitation with high spatial
heterogeneity can dominate the spatial variability of soil moisture and thus affect the spatial variability of
grassland NEE (Wu et al., 2011; Jongen et al., 2011). However, using $0.25°x0.25°$ reanalysis precipitation data
(Zeng et al., 2020) may make it difficult for predictive models to capture this spatial heterogeneity around the
flux station.
Since few of the studies included in this meta-analysis considered the effect of variation in flux footprint, this
feature was difficult to consider in this study. However, its influence should still be further investigated in future
studies. With flux footprints calculated (Kljun et al., 2015) and the factors around the flux site (Walther et al.,
2021) that affect the flux footprint incorporated, .it is promising to clarify this issue.
**4.1.3 Possible unbalance of training and validation sets**
In addition to the time scale of the models, the most significant differences in model accuracy and performance
were found in the heterogeneity within the NEE dataset and the match of the training set and validation set.
Often NEE simulations can achieve high accuracy in local studies, where the main factor negatively affecting
model accuracy may be the interannual variability in the relationship between NEE and covariates. However,
the complexity may increase when the dataset contains a large study area, many sites, PFTs, and year spans.
Under this condition, the accuracy of the model in the 'leave one site out' validation may be more dependent on
the correlation and match between the training and validation sets (Jung et al., 2020). When the model is applied
to an outlier site (of which the NEE, covariates, and their relationship are very different compared with the
remaining sites), it appears to be difficult to achieve a high prediction accuracy (Jung et al., 2020). If we further
upscale the prediction model to large spatial and time scales, the uncertainties involved may be difficult to
assess (Zeng et al., 2020). We can only infer the possible model accuracy based on the similarity of the
distribution of predictors in the predicted grid to that of the existing sites in the model. In the upscaling process,
reanalysis data with coarse spatial resolution are often used as an alternative for site-scale meteorological
predictors. However, most studies did not assess in detail the possible errors associated with spatial mismatches
in this operation.

In summary, the site-scale NEE predictions may require more focus on the internal heterogeneity of the NEE
dataset and the matching of the training set and validation set, and also require a better understanding of the
influence of different scales of the same variable (e.g. site-scale precipitation and grid-scale precipitation in the
reanalysis meteorological data) across modeling and upscaling steps. For the prediction of other carbon fluxes
such as methane fluxes (in the same framework as the NEE predictions), the results of this study may also be
partially applicable, although there may be significant differences in the use of specific predictors (Peltola et al.,

461 2019).

**4.2 Uncertainties**
The uncertainties in this analysis may include:
a) Publication bias and weighting: Publication bias is not refined due to the limitations of the number of
articles that can be included. Meta-analyses often measure the quality of journals and the data availability
(Borenstein et al., 2011; Field and Gillett, 2010) to determine the weighting of the literature in a
comprehensive assessment. However, a high proportion of the articles in this study did not make flux
observations publicly available or share the NEE prediction models developed. Furthermore, meta-analysis
studies in other fields typically measure the impact of papers by evidence/data volume, and the variance of
the evaluated effects (Adams et al., 1997; Don et al., 2011; Liu et al., 2018). However, in this study,
because no convincing method is found to quantify the weights of results from included articles, some
features (e.g. the number of flux sites, the span of years) were directly assessed rather than used to
determine the weights of the articles.
b) Limitations of the criteria for inclusion in the literature: in the model accuracy-based evaluation, we
selected only literature that developed multiple regression models. Potentially valuable information from
univariate regression models was not included. In addition, only papers in high-quality English journals
were included in this study to control for possible errors due to publication bias. However, many studies
that fit this theme may have been published in other languages or other journals.
c) Independence between features: There is dependence between the evaluated features (e.g. the dependency
between the spatial extent and the number of sites). It may negatively affect the assessment of the impact
of individual features on the accuracy of the model, although the BN-based analysis of joint effects can
reduce the impact of this dependence between variables by specifying causal relationships between
features. The interference of unknown dependencies between features may still not be eliminated when we
focus on the effects of an individual feature on the model performance. We should pay more attention to
the effect of features on model accuracy individually in future studies, and it may be valuable to keep other
features as constants while changing the level of only one feature and assessing the difference. It may help
us to understand the real sensitivity of model accuracy to different features in specific conditions. The
sample size collected in this study (178 records in total) is not very large. This also suggests that more
future efforts should be devoted to the comprehensive evaluation and summarization of NEE simulations.

Additionally, there are still other potential factors not considered by this study such as the uncertainty of climate
data (site vs reanalysis), footprint matching between site and satellite images, etc. Overall, although the
quantitative results of this study should be used with caution, they still have positive implications for guiding
future such studies.
**5 Conclusion**
We performed a meta-analysis of the site-scale NEE simulations combining in situ flux observations,
meteorological, biophysical, and ancillary predictors, and machine learning. The impacts of various features
throughout the modeling process on the accuracy of the model were evaluated. The main findings of this study
include:
1.  RF and SVM performed better than other evaluated algorithms.
2.  The impact of time scale on model performance is significant. Models with larger time scales have lower
average R-squared, especially when the time scale exceeds the monthly scale. Models with half-hourly
scales (average R-squared = 0.73) were significantly more accurate than models with daily scales (average
R-squared = 0.5).
3.  Among the commonly used predictors for NEE, there are significant differences in the predictors used and
their impacts on model accuracy for different PFTs.
4.  It is necessary to focus on the potential imbalance between the training and validation sets in NEE
simulations. Studies at continental and global scales (average R-squared = 0.37) with multiple PFTs, more
sites, and a large span of years correspond to lower R-squared than studies at local (average R-squared =
0.69) and regional scales (average R-squared = 0.7).

**Acknowledgments**

We thank the editors and three anonymous referees for their insightful comments on this paper which substantially improved.

**Financial support**

This research was supported by the National Natural Science Foundation of China (Grant No. U1803243), the Key projects of the Natural Science Foundation of Xinjiang Autonomous Region (Grant No. 2022D01D01), the Strategic Priority Research Program of the Chinese Academy of Sciences (Grant No. XDA20060302), and High-End Foreign Experts Project.

**Author contributions**

H.S and G.L initiated this research and were responsible for the integrity of the work as a whole. H.S performed formal analysis, and calculations and drafted the manuscript. H.S, G.L, X.M, X.Y, Y.W, W.Z, M.X, C.Z, and Y.Z were responsible for the data collection and analysis. G.L, P.D.M, T.V.D.V, O.H, and A.K contributed resources and financial support.

**Competing interests**

The authors declare that they have no conflict of interest.

**Data availability**

The data used in this study can be accessed by contacting the first author (shihaiyang16@mails.ucas.ac.cn) based on a reasonable request.

**Code availability**

The code used in this study can be accessed by contacting the first author (shihaiyang16@mails.ucas.ac.cn) based on a reasonable request.

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
