# Peer review of "Exchange Simulations Based on Machine Learning and"

_Biogeosciences, 2022_

## Author Comment (AC1)

**Response to referee comments**

**Referee #1**

In this manuscript, the impacts of features such as the machine learning algorithm, the temporal scales of the observed flux data, and the PFT of the flux sites on the accuracy of the model were evaluated by the authors. The results of this study can provide some general guidance for the selection of feature factors during future NEE simulations. This manuscript logic is clear and well arranged, in terms of criteria for article selection, the choice of analysis methods, and the uncertainties in this analysis of the article. While, there are some problems need to be revised before this manuscript can be published, following is the detailed advices.

Response: We would like to thank the reviewer for the positive comments and the time invested to review our manuscript. The revised manuscript will follow the reviewer's recommendations.

L34, the 2 of $CO_2$ should be subscripted;

We will correct it.

L63, "soil temperature(Ta)" in parentheses should be "Ts";

We will correct it.

L68, It can be expressed as "in models that include multiple PFTs" without writing the full name of the PFT;

We will correct it.

L189-191, please rewrite the sentences. I guess the author wants to express that MLR is weaker than ANN, SVM, and RF because MLR did not divide the training and validations sets. The logic of this sentence is confusing because of the inappropriate use of the words "Unexpectedly" and "not worse than";

Response: Thank you for the insightful comments. We will modify the expression of this sentence more logically.

L230, please add references to support "the lag of precipitation and NDVI/EVI in effect on NEE";

Response: Thank you for the insightful comments. References (Hao et al., 2010; Cranko Page et al., 2022) will be added to support this description.

L431-434, the reference title formatting is inconsistent with others;

We will check and correct it.

Figure 3, there is no scale bar and north arrow;

We will add a scale bar and north arrow to the map.

The 2 in $R^2$ needs to be superscripted in all figures throughout the manuscript, e.g. Figure 5, Figure 6, Figure 7, Figure 8, etc.

We will check and correct them which should be superscripted.

**References**

Cranko Page, J., De Kauwe, M. G., Abramowitz, G., Cleverly, J., Hinko-Najera, N., Hovenden, M. J., Liu, Y., Pitman, A. J., and Ogle, K.: Examining the role of environmental memory in the predictability of carbon and water fluxes across Australian ecosystems, 19, 1913–1932, https://doi.org/10.5194/bg-19-1913-2022, 2022.

Hao, Y., Wang, Y., Mei, X., and Cui, X.: The response of ecosystem $CO_2$ exchange to small precipitation pulses over a temperate steppe, Plant Ecol, 209, 335–347, https://doi.org/10.1007/s11258-010-9766-1, 2010.

---

## Author Comment (AC3)

**Response to referee comments**

**Referee #3**

This manuscript conducts a meta-analysis to explore the uncertainty of simulating NEE using comparing machine learning techniques, time-scale and spatial-scale changes, and input variables. This is an important topic to solve the difference between observed and predicted NEE. However, this manuscript doesn't clarify the objectives and detail of data processing. Oversimple descriptions in the Methods section makes readers confusing. Additionally, the usage of too many speculative explanations in the discussion section is hard to draw universal conclusions. This manuscript doesn't clarify the motivation of the work, especially in the advantages and potential of ML. In summary, the paper needs to be substantially revised, and some parts need further elaboration.

Response: We would like to thank the reviewer for the positive comments and the time invested to review our manuscript. The revised manuscript will follow the reviewer's recommendations.

L32, This sentence hardly reflects the scientific value of this paper.

Response: Thank you for the insightful comments. We will delete this sentence.

L40-L43, The advantages and the current situation applied to ML need to be further reviewed, which is beneficial for readers to understand the purpose of introducing ML in this paper.

Response: Thank you for the insightful comments. We will further state the advantages and state-of-the-art of using machine learning for NEE simulations compared to process-based models. Previous process-based and empirical models are limited in their performance in NEE prediction due to our poor understanding of the mechanisms of NEE. Compared to process-based models, machine learning-based NEE prediction models can improve accuracy by establishing complex relationships between observed NEE and environmental variables in a data-driven manner.

L45, The sentence, "a synthesis evaluation is …limited", needs to be further explained otherwise it is hardly understood.

Response: We will provide more descriptive text, such as the limitations of the existing local multi-model evaluation.

L49-50, need references, preferably with 2 examples

Response: We will provide the appropriate references here.

L52-54, There is a logical gap between this sentence and the previous statements.

Response: We will emphasize the importance of evaluation studies from 'local' to 'global', thus making the logic smooth.

L88-93, The uncertainty caused by spatio-temporal heterogeneity cannot be confused with the volume of data sets. Because large-data volume does not equate to higher heterogeneity. Big data provides more opportunities to build balanced-training data. This section may need to be rewritten.

Response: Thank you for the insightful comments. Indeed, data volume does not always correspond to large spatiotemporal heterogeneity. For example, a site with a long year span and small interannual climate variability may have a large data volume, but it may not contain large spatiotemporal heterogeneity. We will rewrite this paragraph to avoid this confusion.

L107-108, need references

Response: We will provide the reference (Marcot and Hanea, 2021) here.

L116, "Other Features" needs to be clarified. The purpose of this manuscript may be to explore: the uncertainty of NEE evaluation results caused by ML techniques, spatio-temporal resolution remote sensing data, and verification methods according to the introduction?

Response: We will revise it as 'machine learning algorithms, Spatio-temporal resolution of remote sensing data, and validation methods'.

L144, An oversimplified description of the workflow, please give an overview and detailed sub-steps of data processing and simulation. It is hard to know the objectives of each analysis for readers.

Response: We will provide more details on data extraction.

L150, Abbreviations in the figure need to be clarified

Response: We will provide the meaning of these abbreviations in the figure caption.

L178, Need scale bar and north arrow in figure 3

Response: We will add a scale bar and north arrow.

L198, It is difficult to find the differences among algorithms using simple comparisons in figure 5a, and needs more statistically testing. Additionally, this

figure confuses me. Why are MLR, RF, SVM, and ANN separately compared? Please provide explanations. Why is PLSR with high R2 removed? Finally, there are also some problems with the image. The caption does not explain the details of the box. Does the line in the box represent the mean or the median?

Response: Thank you for the insightful comments. MLR, RF, SVM, and ANN are the more commonly used methods. Other algorithms such as PLSR may have too small a sample size. Since cross-study comparisons of algorithm accuracy include differences in data used in model construction, we perform a pairwise comparison of these four algorithms in figure 5b. Multiple models are developed for uniform training data in a single study so that the interference of data variation is removed. The line in the box shows the median. We will revise this figure (and the caption of the figure) in detail to provide more clarification.

L205, Avoid using the word "significant" without statistically testing

Response: We will replace the word 'significant'.

L206-210, It is hard to read the trend in Figure 6. Recommend adding a line chart to demonstrate the decreasing trend.

Response: We will add trend lines.

L212, There are no details of the boxplot. Are all models incorporated into the time-scales comparison, or only RF, SVM, and ANN? Please add the details of data processing.

Response: All models have been included in the assessment of time-scales variations.

L223, Also, use these words carefully without statistically testing.

Response: We will replace the word 'significantly'.

L263, Need to reorder the y-axis text in figure 8. Furthermore, a serious question is whether the comparison analysis of these variables keeps other variables constant? If not, conclusions based on comparisons of R2 may not hold water.

Response: We will readjust the order. Indeed, in the assessment of the impacts of variables, the interference between variables is not eliminated (and indeed it is difficult to keep the other variables constant). Therefore, the subsequent Bayesian network-based analysis can be considered a multivariate analysis with the elimination of the interference between the variables.

L299, Lacking the in-depth discussion of the uncertainty of NEE prediction resulting from time-scale change.

Response: We will improve the discussion section by discussing in more depth the impact of factors affecting NEE prediction (especially the change in the time scale you mentioned). Since this study only provides findings based on statistics obtained, we will compare some of the explanations of possible effects on time scales at the mechanistic level in previous studies.

L308, There are too many speculative parts and insufficient supporting materials in section 4.1 of discussed.

Response: We will add more references to support our discussion in section 4.1.

For example, we will add three references for the following discussion text:

'In addition, the impacts of lagged effects (Hao et al., 2010; Cranko Page et al., 2022) of covariates are not considered in most models, which may underestimate the degree of explanation of NEE for some predictor variables (e.g. precipitation). Most of the machine learning-based models use only the average Ta and do not take into account the maximum temperature, minimum temperature, daily difference in temperature, etc., as in the process-based ecological models (Mitchell et al., 2009).'

L321-323, The discussion of model accuracy difference caused by satellites needs careful. This sentence needs further support. Are you implying that the time scale compensates for the uncertainty caused by the spatial scale?

Response: The discussion on this in the current version was not careful although it is true that the temporal availability of MODIS data and Landsat data differs greatly. We will consider adding references.

L326-330, This sentence is too long

Response: We will simplify this sentence.

L330-332, The time-scale discussion containing spatial-scale matching will confuse readers.

Response: Thank you for the insightful comments. We will separate the discussion of time-scale and the discussion of spatial-scale matching. In the current version, these two parts are placed in one paragraph and it may confuse readers. We will revise this.

L349, Does "coarse-resolution" here note spatial resolution or temporal resolution?

Response: This refers to spatial resolution. We will clarify this in the manuscript.

**References**

Cranko Page, J., De Kauwe, M. G., Abramowitz, G., Cleverly, J., Hinko-Najera, N., Hovenden, M. J., Liu, Y., Pitman, A. J., and Ogle, K.: Examining the role of environmental memory in the predictability of carbon and water fluxes across Australian ecosystems, 19, 1913–1932, https://doi.org/10.5194/bg-19-1913-2022, 2022.

Hao, Y., Wang, Y., Mei, X., and Cui, X.: The response of ecosystem $CO_2$ exchange to small precipitation pulses over a temperate steppe, Plant Ecol, 209, 335–347, https://doi.org/10.1007/s11258-010-9766-1, 2010.

Marcot, B. G. and Hanea, A. M.: What is an optimal value of k in k-fold cross-validation in discrete Bayesian network analysis?, Comput Stat, 36, 2009–2031, https://doi.org/10.1007/s00180-020-00999-9, 2021.

Mitchell, S., Beven, K., and Freer, J.: Multiple sources of predictive uncertainty in modeled estimates of net ecosystem $CO_2$ exchange, Ecological Modelling, 220, 3259–3270, https://doi.org/10.1016/j.ecolmodel.2009.08.021, 2009.

---

## Author Response (AR1)

**Response to referee comments and actions**

**Referee #1**

In this manuscript, the impacts of features such as the machine learning algorithm, the temporal scales of the observed flux data, and the PFT of the flux sites on the accuracy of the model were evaluated by the authors. The results of this study can provide some general guidance for the selection of feature factors during future NEE simulations. This manuscript logic is clear and well arranged, in terms of criteria for article selection, the choice of analysis methods, and the uncertainties in this analysis of the article. While, there are some problems need to be revised before this manuscript can be published, following is the detailed advices.
Response: We would like to thank the reviewer for the positive comments and the time invested to review our manuscript. The revised manuscript will follow the reviewer's recommendations.

L34, the 2 of $CO_2$ should be subscripted;
We will correct it.
Action: Corrected.

L63, "soil temperature(Ta)" in parentheses should be "Ts";
We will correct it.
Action: Corrected.

L68, It can be expressed as "in models that include multiple PFTs" without writing the full name of the PFT;
We will correct it.
Action: Corrected.

L189-191, please rewrite the sentences. I guess the author wants to express that MLR is weaker than ANN, SVM, and RF because MLR did not divide the training and validations sets. The logic of this sentence is confusing because of the inappropriate use of the words "Unexpectedly" and "not worse than";
Response: Thank you for the insightful comments. We will modify the expression of this sentence more logically.
Action: This sentence was deleted in the revised manuscript.

L230, please add references to support "the lag of precipitation and NDVI/EVI in effect on NEE";
Response: Thank you for the insightful comments. References (Hao et al., 2010; Cranko Page et al., 2022) will be added to support this description.
Action: References added.

L431-434, the reference title formatting is inconsistent with others;

We will check and correct it.

Action: Corrected.

Figure 3, there is no scale bar and north arrow;

We will add a scale bar and north arrow to the map.

Action: Scale bar and north arrow added.

The 2 in R2 needs to be superscripted in all figures throughout the manuscript, e.g. Figure 5, Figure 6, Figure 7, Figure 8, etc.

We will check and correct them which should be superscripted.

Action: Corrected (superscripted).

**Response to referee comments and actions**

**Referee #2**

This study implemented a meta-analysis of current NEE prediction studies. Overall, the topic is interesting and the methodology is innovative, as few researchers in past studies have used R2 or other accuracy metrics to compare models of different studies. Although the number of available models is not large, some of the findings of this study have adding-values and implications at the cross-study (different focus, data, models) level. This manuscript is of interest to BG readers (especially researchers using machine learning to predict NEE). The following issues should be clarified before acceptance.

Response: We would like to thank the reviewer for the positive comments and the time invested to review our manuscript. The revised manuscript will follow the reviewer's recommendations.

**Main comments:**

The authors have already mentioned the inconsistency between the area of the flux footprint and the area extracted from remote sensing data (e.g. 2x2 km). So, could the authors extract this information from the literature and further analyze this effect? I believe this analysis will be interesting.

Response: Thank you for the insightful comments. Indeed the scale of the explanatory variables affects how well they match the scale of the flux observations. We will consider discussing this issue more in-depth or extracting this information from the literature and further evaluate this effect in various PFTs.

Action: Elaborated in the discussion section: 'The effects of this mismatch between explanatory variables and flux footprints may be diverse for different PFTs. For example, for cropland types, the NEE is monitored at a range of several hundred meters around the flux towers, but remote sensing variables such as FAPAR, NDVI, LAI, etc. can be extracted at coarse scales (e.g., 2x2 km), some effects outside the extent of the flux footprint (Chu et al., 2021; Walther et al., 2021) are incorporated (e.g., planting structures with high spatial heterogeneity, agricultural practices such as irrigation). And for more homogeneous types such as grasslands, coarse-scale meteorological data may still cause spatial mismatches, even though the differences in land cover types within the 2x2 km and 200x200 m extent around the flux stations in grasslands may not be considerable. For example, precipitation with high spatial heterogeneity can dominate the spatial variability of soil moisture and thus affect the spatial variability of grassland NEE (Wu et al., 2011; Jongen et al., 2011). However, using 0.25°x0.25° reanalysis precipitation data (Zeng et al., 2020) may make it difficult for predictive models to capture this spatial heterogeneity around the flux station.' **(Line 402)**

The discussion section is not in-depth enough. The authors should adequately compare the differences between some conclusions in previous studies and the findings of this manuscript.

Response: Thank you for the insightful comments. We will further improve the discussion section of this manuscript by incorporating/comparing findings from previous literature (e.g., the study of uncertainty in modeling practices in some local studies).

Action: Elaborated in the discussion section: 'Compared to previous comparisons of machine learning-based NEE prediction models, this study is more comprehensive. Previous studies (Abbasian et al., 2022) have also found advantages of RF over other algorithms in NEE prediction. This study consolidated this finding using a larger amount of evidence. Previous studies (Tramontana et al., 2016) have also compared the impact of different practices in NEE prediction models based on the R-squared, such as comparing the difference in accuracy between the two predictor combinations (i.e., using only remotely sensed data and using remotely sensed data and meteorological data together). In contrast, since this study incorporated more detailed factors influencing model accuracy, the comprehensiveness of the understanding of such issues was deepened. However, there are still many uncertainties and challenges in NEE prediction not clarified in this study.' **(Line 345)**

**Other comments:**

In Table 1, GPP is also used as a keyword in the literature collection? Clarify.

Response: Our inclusion of the keyword GPP was to ensure that as much of the literature as possible was included because NEE was predicted along with GPP in some literature.

In Table 2, evapotranspiration (ET) is also used as a predictor. Is ET here the latent heat observed by the flux station? Clarify.

Response: We would modify it to 'evapotranspiration (ET) as the latent heat observed by the flux station'.

Action: Modified as 'evapotranspiration (ET) (i.e., the latent heat observed by the flux station)'

In Figure 8, the categories should be reordered.

Response: We will modify the order.

Action: The categories were reordered.

The area observed by the flux station should be larger than 100 x 100 m (usually a few hundred meters).

Response: We will modify it to 'a few hundred meters'. The observation extent of the flux footprint is influenced by many factors such as wind speed and therefore varies within a few hundred meters.

Action: modified to 'a few hundred meters'.

**Response to referee comments and actions**

**Referee #3**

This manuscript conducts a meta-analysis to explore the uncertainty of simulating NEE using comparing machine learning techniques, time-scale and spatial-scale changes, and input variables. This is an important topic to solve the difference between observed and predicted NEE. However, this manuscript doesn't clarify the objectives and detail of data processing. Oversimple descriptions in the Methods section makes readers confusing. Additionally, the usage of too many speculative explanations in the discussion section is hard to draw universal conclusions. This manuscript doesn't clarify the motivation of the work, especially in the advantages and potential of ML. In summary, the paper needs to be substantially revised, and some parts need further elaboration.

Response: We would like to thank the reviewer for the positive comments and the time invested to review our manuscript. The revised manuscript will follow the reviewer's recommendations.

L32, This sentence hardly reflects the scientific value of this paper.

Response: Thank you for the insightful comments. We will delete this sentence.

Action: Deleted.

L40-L43, The advantages and the current situation applied to ML need to be further reviewed, which is beneficial for readers to understand the purpose of introducing ML in this paper.

Response: Thank you for the insightful comments. We will further state the advantages and state-of-the-art of using machine learning for NEE simulations compared to process-based models. Previous process-based and empirical models are limited in their performance in NEE prediction due to our poor understanding of the mechanisms of NEE. Compared to process-based models, machine learning-based NEE prediction models can improve accuracy by establishing complex relationships between observed NEE and environmental variables in a data-driven manner.

Action: Elaborated 'Many researchers have tried to use a data-driven approach as an alternative (Fu et al., 2014; Tian et al., 2017; Tramontana et al., 2016; Jung et al., 2011). On the one hand, it was made possible by the increase in the growth of global carbon flux observations and the large amount of flux observation data being accumulated. Since the 1990s, the use of the eddy covariance technique to monitor NEE has been rapidly promoted (Baldocchi, 2003). Several regional and global flux measurement networks have been established for the big data management of the flux sites, including CarboEuro-flux (Europe), AmeriFlux (North America), OzFlux (Australia), ChinaFlux (China), FLUXNET (global), etc. On the other hand, machine learning approaches are

increasingly used to extract patterns and insights from the ever-increasing stream of geospatial data (Reichstein et al., 2019). The rapid development of various algorithms and high public availability of model tools in the field of machine learning have made these techniques easily available to more researchers in the field of geography and ecology (Reichstein et al., 2019). Since the above two major advances in the last two decades, various machine learning algorithms have been used to simulate NEE at the flux station scale with various predictor variables (e.g., meteorological variables, biophysical variables) incorporated for spatial and temporal mapping of NEE or understanding the driving mechanisms of NEE.' **(line 37)**

L45, The sentence, "a synthesis evaluation is …limited", needs to be further explained otherwise it is hardly understood.

Response: We will provide more descriptive text, such as the limitations of the existing local multi-model evaluation.

Action: revised as 'To date, studies on using machine learning to predict NEE have a high diversity in terms of modeling approaches. To obtain a comprehensive understanding of machine learning-based NEE prediction, a synthesis evaluation of these machine learning models is necessary.' **(line 53)**

L49-50, need references, preferably with 2 examples

Response: We will provide the appropriate references here.

Action: revised as 'Many studies have demonstrated the effectiveness of their proposed improvements (i.e., using predictors with a higher spatial resolution (Reitz et al., 2021) and using data from the local flux site network (Cho et al., 2021)) by comparing with previous studies.'   **(Line 58)**

L52-54, There is a logical gap between this sentence and the previous statements.

Response: We will emphasize the importance of evaluation studies from 'local' to 'global', thus making the logic smooth.

Action: revised as 'We are more interested in guidelines with universal applicability that improve the model accuracy, such as the selection of appropriate predictors and algorithms under different conditions. Therefore, we should synthesize the results of models applied to different conditions and regions to obtain general insights.' **(line 63)**

L88-93, The uncertainty caused by spatio-temporal heterogeneity cannot be confused with the volume of data sets. Because large-data volume does not equate to higher heterogeneity. Big data provides more opportunities to build balanced-training data. This section may need to be rewritten.

Response: Thank you for the insightful comments. Indeed, data volume does not always correspond to large spatiotemporal heterogeneity. For example, a site with a long year span and small interannual climate variability may have a large data volume, but it may not contain large spatiotemporal heterogeneity. We will rewrite this paragraph to avoid this confusion.

Action: The text concerning on the volume of data was removed: '(b) The spatio-temporal heterogeneity of data sets, and validation method: The spatio-temporal heterogeneity of the dataset may affect model accuracy.' **(line 98)**

L107-108, need references

Response: We will provide the reference (Marcot and Hanea, 2021) here.

Action: Reference (Marcot and Hanea, 2021) added.

L116, "Other Features" needs to be clarified. The purpose of this manuscript may be to explore: the uncertainty of NEE evaluation results caused by ML techniques, spatio-temporal resolution remote sensing data, and verification methods according to the introduction?

Response: We will revise it as 'machine learning algorithms, Spatio-temporal resolution of remote sensing data, and validation methods'.

Action: Revised as 'In this study, to evaluate the impacts of predictors use, algorithms, spatial/temporal scale, and validation methods on model accuracy, we performed a meta-analysis of papers with prediction models that combine NEE observations from flux towers, various predictors, and machine learning for the data-driven NEE simulations.' (line 126)

L144, An oversimplified description of the workflow, please give an overview and detailed sub-steps of data processing and simulation. It is hard to know the objectives of each analysis for readers.

Response: We will provide more details on data extraction.

Action: Elaborated as:

'The information of R-squared (at the validation phase) and the associated model features reported in the article are considered as one data record for the formal meta-analysis (i.e., each R-squared record corresponded to a prediction model). From the included papers, R-squared records and various features (Table 2) involved in the NEE modeling framework (Fig. 2) were extracted (including the used algorithms, modeling/validation methods, remote sensing data, meteorological data, biophysical data and ancillary data). In some studies, multiple algorithms were applied to the same dataset, or models with

different features were developed. In these cases, multiple data records will be documented.

In the practical information extracting step, we categorized such features in a comparable manner. First, we categorized the various algorithms used in these papers, although the same algorithm may also have a variant form or an optimized parameter scheme. They are categorized into the following families of algorithms: Random Forests (RF), Multiple Linear Regressions (MLR), Artificial Neural Networks (ANN), Support Vector Machines (SVM), Partial Least Squares Regression (PLSR), Generalized additive model (GAM), Boosted Regression Tree (BRT), Bayesian Additive Regression Trees (BART), Cubist , model tree ensembles (MTE). Second, we classified the spatial scales of these studies. Models with study areas (spatial extent covered by flux stations) smaller than 100x100 km were classified as 'local' scale models, those with study area sizes exceeding continental scale were classified as 'global' scale, and those with study area sizes in between were classified as 'regional' scale. Third, for various predictors, we only recorded whether the predictors were used or not without distinguishing the detailed data sources and categories (e.g., grid meteorological data from various reanalysis datasets and in-situ meteorological observations from flux stations), measurement methods (e.g., soil moisture measured/estimated by remote sensing or in situ sensors), etc. Fourth, we documented PFTs for the prediction models from the description of study area or sites in these papers. They are classified into the following types: forest, grassland, cropland, wetland, savannah, tundra, and multi-PFTs (models containing a mixture of multiple PFTs). Models not belonging to the above PFTs were not given a PFT field and were not included in the subsequent analysis on the PFT differences. Other features (Table 2) are extracted directly from the corresponding descriptions in the papers in an explicit manner.' **(line 152-177)**

L150, Abbreviations in the figure need to be clarified

Response: We will provide the meaning of these abbreviations in the figure caption.

Action: The figure caption is elaborated: 'Prec, Ta, Rn, Ws, RH, and VPD represent precipitation, air temperature, net surface radiation, wind speed, relative humidity, and vapour-pressure deficit. FAPAR is the fraction of absorbed photosynthetically active radiation. LST is the land surface temperature. LAI ia the leaf area index.'

L178, Need scale bar and north arrow in figure 3

Response: We will add a scale bar and north arrow.

Action: Added.

L198, It is difficult to find the differences among algorithms using simple comparisons in figure 5a, and needs more statistically testing. Additionally, this figure confuses me. Why are MLR, RF, SVM, and ANN separately compared? Please provide explanations. Why is PLSR with high R2 removed? Finally, there are also some problems with the image. The caption does not explain the details of the box. Does the line in the box represent the mean or the median?

Response: Thank you for the insightful comments. MLR, RF, SVM, and ANN are the more commonly used methods. Other algorithms such as PLSR may have too small a sample size. Since cross-study comparisons of algorithm accuracy include differences in data used in model construction, we perform a pairwise comparison of these four algorithms in figure 5b. Multiple models are developed for uniform training data in a single study so that the interference of data variation is removed. The line in the box shows the median. We will revise this figure (and the caption of the figure) in detail to provide more clarification.

Action: Revised as 'Among the more frequently used algorithms, ANN and SVM performed better (Fig. 5a) on average across studies (lightly better than RF). On the other hand, since cross-study comparisons of algorithm accuracy include differences in data used in model construction, we performed a pairwise comparison (Fig. 5b) of these four algorithms (i.e., ANN, SVM, RF, and MLR). In these studies, multiple models are developed for consistent training data with the interference of training data differences removed. It shows that RF and SVM perform best in the inter-study comparison (Fig. 5b). Whereas ANN performed slightly worse than RF and SVM, all three of them were stronger than MLR. Overall, the performance of RF and SVM may be good and similar in the NEE simulations.' **(line 223)**

To explain the meaning of the lines in the boxplot, the figure caption is elaborated: 'In the panel (a), the horizontal line in the box indicates the medians. The top and bottom border lines of the box indicate the 75% and 25% percentiles, respectively.

[Figure]

Figure 5. Differences in model accuracy (R-squared) using different algorithms across studies (a) and internal comparisons of the model accuracy (R-squared) of selected pairs of algorithms within

individual studies (b). Regression algorithms: Random Forests (RF), Multiple Linear Regressions (MLR), Artificial Neural Networks (ANN), Support Vector Machines (SVM), Partial Least Squares Regression (PLSR), Generalized additive model (GAM), Boosted Regression Tree (BRT), Bayesian Additive Regression Trees (BART), Cubist, model tree ensembles (MTE). In panel (a), the horizontal line in the box indicates the medians. The top and bottom border lines of the box indicate the 75% and 25% percentiles, respectively.

L205, Avoid using the word "significant" without statistically testing

Response: We will replace the word 'significant'.

Action: Replaced with 'considerable'.

L206-210, It is hard to read the trend in Figure 6. Recommend adding a line chart to demonstrate the decreasing trend.

Response: We will add trend lines.

Action: A trend line subplot figure was added as the regression line of values of R-squared and time scales (in days) in Figure 6.

[Figure]

Figure 6. Differences in model accuracy (R-squared) at different time scales across studies with the linear regression between R-squared and time scales (a), and comparison of the model accuracy (R-squared) of selected pairs of time scales within individual studies (b). All model records were included in the panel (a), while studies that used multiple time scales (with other model characteristics unchanged) were included in the panel (b). Time scales: 0.02 days (half-hourly), 0.04 days (hourly), 30 days (monthly), 90 days (quarterly).

L212, There are no details of the boxplot. Are all models incorporated into the time-scales comparison, or only RF, SVM, and ANN? Please add the details of data processing.

Response: All models have been included in the assessment of time-scales variations.

Action: Added in the figure caption: 'All model records were included in the panel (a), while studies that used multiple time scales (with other model characteristics unchanged) were included in the panel (b).'

L223, Also, use these words carefully without statistically testing.

Response: We will replace the word 'significantly'.

Action: 'significantly' deleted here.

L263, Need to reorder the y-axis text in figure 8. Furthermore, a serious question is whether the comparison analysis of these variables keeps other variables constant? If not, conclusions based on comparisons of R2 may not hold water.

Response: We will readjust the order. Indeed, in the assessment of the impacts of variables, the interference between variables is not eliminated (and indeed it is difficult to keep the other variables constant). Therefore, the subsequent Bayesian network-based analysis can be considered a multivariate analysis with the elimination of the interference between the variables.

Action: The y-axis has been reordered.

The uncertainties and limitations of this analysis due to the joint control of multiple features on the R-squared and also the covariance between some of the features are described in the Discussion section:

'There is dependence between the evaluated features (e.g. the dependency between the spatial extent and the number of sites). It may negatively affect the assessment of the impact of individual features on the accuracy of the model, although the BN-based analysis of joint effects can reduce the impact of this dependence between variables by artificially specifying causal relationships between features. The interference of unknown dependencies between features may still not be eliminated when we focus on the effects of an individual feature on the model performance. The sample size collected in this study (178 records in total) is not very large. This also suggests that more future efforts should be devoted to the comprehensive evaluation and summarization of NEE simulations.' **(line 460)**

L299, Lacking the in-depth discussion of the uncertainty of NEE prediction resulting from time-scale change.

Response: We will improve the discussion section by discussing in more depth the impact of factors affecting NEE prediction (especially the change in the time scale you mentioned). Since this study only provides findings based on statistics obtained, we will compare some of the explanations of possible effects on time scales at the mechanistic level in previous studies.

Action: elaborated as:

'Previous studies have reported the dependence of the NEE variability and mechanism on the time scales. On the one hand, the importance of variables affecting NEE varies at different time scales. For example, in tropical and subtropical forests in southern China (Yan et al., 2013), seasonal NEE variability is predominantly controlled by soil temperature and moisture, while interannual NEE variability is controlled by the annual precipitation variation. A study (Jung et al., 2017) showed that for annual-scale NEE variability, water availability and temperature were the dominant drivers at the local and global scales, respectively. This indicates the need to recognize the temporal and spatial driving mechanisms of NEE in advance in the development of NEE prediction models. On the other hand, dependence may exist between NEE anomalies at various time scales. For example, previous studies (Luyssaert et al., 2007) showed that short-term temperature anomalies may interpret both the daily and seasonal NEE anomalies. This implies that the models at different time scales may not be independent. In the previous studies, the relationship between prediction models at different scales has not been well investigated, and it may be valuable to compare the relations between data and models at different scales in depth. Larger time scales correspond to lower model accuracy, possibly related to the fact that some small-time-scale relations between NEE and covariates (especially meteorological variables) are smoothed. In particular, for models with time scales smaller than one day (e.g. half-hourly models), the 8-daily and 16-daily biophysical variable data obtained from satellite remote sensing are difficult to explain the temporal variation in the sub-daily NEE. Therefore, for models at small time scales (i.e. half-hourly, hourly, daily scale models), in situ meteorological variables may be more important. The inclusion of some ancillary variables (e.g. soil texture, topographic variables) with no temporal dynamic information may be ineffective unless many sites are included in the model and the spatial variability of the ancillary variables for these sites is sufficiently large (Virkkala et al., 2021).

In terms of completeness and purity of training data, hourly and daily models can be better compared to monthly and yearly models. Hourly and daily models can usually preclude those low-quality data and gaps in the flux observations. However, for monthly and yearly scale models, gap-filling (Ruppert et al., 2006; Moffat et al., 2007; Zhu et al., 2022) is necessary because there are few complete and continuous flux observations without data gaps on the monthly to yearly scales. Since various gap-filling techniques rely on environmental factors (Moffat et al., 2007) such as meteorological observations, this may introduce uncertainty in the predictive models (i.e., a small fraction of the observed information of NEE is estimated from a combination of independent variables). How it

would affect the accuracy of prediction models at various time scales remains uncertain, although various gap-filling techniques have been widely used in the pre-processing of training data.' **(Line 358-388)**

L308, There are too many speculative parts and insufficient supporting materials in section 4.1 of discussed.

Response: We will add more references to support our discussion in section 4.1.

For example, we will add three references for the following discussion text:

'In addition, the impacts of lagged effects (Hao et al., 2010; Cranko Page et al., 2022) of covariates are not considered in most models, which may underestimate the degree of explanation of NEE for some predictor variables (e.g. precipitation). Most of the machine learning-based models use only the average Ta and do not take into account the maximum temperature, minimum temperature, daily difference in temperature, etc., as in the process-based ecological models (Mitchell et al., 2009).'

Action: We have rearranged the structure of 4.1 (into 4.1.1, 4.1.2, and 4.1.3), and many references have been added to support what is discussed.

In addition, we have removed some speculative text.

L321-323, The discussion of model accuracy difference caused by satellites needs careful. This sentence needs further support. Are you implying that the time scale compensates for the uncertainty caused by the spatial scale?

Response: The discussion on this in the current version was not careful although it is true that the temporal availability of MODIS data and Landsat data differs greatly. We will consider adding references.

Action: We removed the discussion text of the impact of the temporal and spatial scales of satellite data, because we recognize that to perform an in-depth analysis of this issue, it may not be sufficient based on only the evidence presented in this study alone.

L326-330, This sentence is too long

Response: We will simplify this sentence.

Action: revised as: 'Since few of the studies included in this meta-analysis considered the effect of variation in flux footprint, this feature was difficult to consider in this study. However, its influence should still be further investigated in future studies. With flux footprints calculated (Kljun et al., 2015) and the factors around the flux site (Walther et al.,

2021) that affect the flux footprint incorporated, .it is promising to clarify this issue.' **(Line 415)**

L330-332, The time-scale discussion containing spatial-scale matching will confuse readers.

Response: Thank you for the insightful comments. We will separate the discussion of time-scale and the discussion of spatial-scale matching. In the current version, these two parts are placed in one paragraph and it may confuse readers. We will revise this.

Action: It has been separated into two sub-sections: '**4.1.1 Variations in time scales**' and '**4.1.2 Scale mismatch of explanatory predictors and flux footprints**'.

L349, Does "coarse-resolution" here note spatial resolution or temporal resolution?

Response: This refers to spatial resolution. We will clarify this in the manuscript.
Action: revised as 'reanalysis meteorological data with coarse **spatial** resolution'

[revised manuscript text omitted]

---

## Author Response (AR2)

**Response to Referee #3**

I observe that the authors paid much attention to my previous comments and did a great job to integrate my remarks into the new version of the manuscript. The majority of my concerns have been addressed. There are still a few minor improvements required. Overall, the manuscript improved considerably. I suggest it for publication after minor revisions.

Response: We would like to thank the reviewer for the positive comments and the time invested to review our manuscript again. The revised manuscript will follow the reviewer's recommendations.

Specific comments:
1. Please add the description of the workflow (Fig. 2) to clarify the main steps in the 2.2 section.
Response: the description of the workflow is clarified in the 2.2. section: 'Typically, the flow of the NEE prediction modeling framework (Fig. 2) based on flux observations and machine learning is as follows: first, half-hourly scale NEE flux observations are aggregated into various time scale NEE data, and gap-filling techniques (Moffat et al., 2007) are often used in this step to obtain complete NEE series when data are missing. Various predictors including meteorological variables, remote sensing-based biophysical variables, etc. are extracted to match site-scale NEE series to generate a training dataset containing the target variable NEE and various covariates. Subsequently, various algorithms are used for the NEE prediction model construction and validated in different ways (e.g., leave-one-site-out validation (Zeng et al., 2020)). Finally, in some studies, prediction models were applied on gridded covariate data to map the regional or global-scale NEE spatial and temporal variations (Zeng et al., 2020; Papale and Valentini, 2003; Jung et al., 2020). The information of R-squared (at the validation phase) and the associated model features reported in the article are considered as one data record for the formal meta-analysis (i.e., each R-squared record corresponding to a prediction model). From the included papers, R-squared records and various features (Table 2) involved in the NEE modeling framework (Fig. 2) were extracted (including the used algorithms, modeling/validation methods, remote sensing data, meteorological data, biophysical data, and ancillary data). In some studies, multiple algorithms were applied to the same dataset, or models with different features were developed (Virkkala et al., 2021; Zhang et al., 2021; Cleverly et al., 2020; Tramontana et al., 2016). In these cases, multiple data records will be documented.' (Line 151)

2. The R2-based comparison with keeping other variables constant may cause potential uncertainty. Need to add statements and instructions in the Discussion section.
Response: elaborated in the Discussion section as 'We should pay more attention to the effect of features on model accuracy individually in future studies, and it may be valuable to keep other features as constants while changing the level of only one feature and assessing the difference. It may help us to understand the real sensitivity of model accuracy to different features in specific conditions.' (Line 480)

3. Please add an explanation for the reason why MLR, RF, SVM, and ANN are separately compared instead of all models and why PLSR with high R2 is removed in the Methodology

section. This response is similar to the response for L198.

Response: elaborated as 'Subsequently, the model accuracies corresponding to different levels of various features are compared in a cross-study fashion. In the evaluation of algorithms and time scales, we also implement comparisons within individual studies. For example, in the evaluation of the effects of the algorithms, we compare the accuracy of models using the same training data and keeping other features as constants in individual studies. In this intra-study comparison step, only algorithms with relatively large sample sizes in the cross-study comparisons were selected.' (Line 188)

**References**

Cleverly, J., Vote, C., Isaac, P., Ewenz, C., Harahap, M., Beringer, J., Campbell, D. I., Daly, E., Eamus, D., He, L., Hunt, J., Grace, P., Hutley, L. B., Laubach, J., McCaskill, M., Rowlings, D., Rutledge Jonker, S., Schipper, L. A., Schroder, I., Teodosio, B., Yu, Q., Ward, P. R., Walker, J. P., Webb, J. A., and Grover, S. P. P.: Carbon, water and energy fluxes in agricultural systems of Australia and New Zealand, 287, https://doi.org/10.1016/j.agrformet.2020.107934, 2020.

Jung, M., Schwalm, C., Migliavacca, M., Walther, S., Camps-Valls, G., Koirala, S., Anthoni, P., Besnard, S., Bodesheim, P., Carvalhais, N., Chevallier, F., Gans, F., S Goll, D., Haverd, V., Köhler, P., Ichii, K., K Jain, A., Liu, J., Lombardozzi, D., E M S Nabel, J., A Nelson, J., O'Sullivan, M., Pallandt, M., Papale, D., Peters, W., Pongratz, J., Rödenbeck, C., Sitch, S., Tramontana, G., Walker, A., Weber, U., and Reichstein, M.: Scaling carbon fluxes from eddy covariance sites to globe: Synthesis and evaluation of the FLUXCOM approach, 17, 1343–1365, https://doi.org/10.5194/bg-17-1343-2020, 2020.

Moffat, A. M., Papale, D., Reichstein, M., Hollinger, D. Y., Richardson, A. D., Barr, A. G., Beckstein, C., Braswell, B. H., Churkina, G., Desai, A. R., Falge, E., Gove, J. H., Heimann, M., Hui, D., Jarvis, A. J., Kattge, J., Noormets, A., and Stauch, V. J.: Comprehensive comparison of gap-filling techniques for eddy covariance net carbon fluxes, 147, 209–232, https://doi.org/10.1016/j.agrformet.2007.08.011, 2007.

Papale, D. and Valentini, R.: A new assessment of European forests carbon exchanges by eddy fluxes and artificial neural network spatialization, 9, 525–535, https://doi.org/10.1046/j.1365-2486.2003.00609.x, 2003.

Tramontana, G., Jung, M., Schwalm, C. R., Ichii, K., Camps-Valls, G., Ráduly, B., Reichstein, M., Arain, M. A., Cescatti, A., Kiely, G., Merbold, L., Serrano-Ortiz, P., Sickert, S., Wolf, S., and Papale, D.: Predicting carbon dioxide and energy fluxes across global FLUXNET sites with regression algorithms, Biogeosciences, 13, 4291–4313, https://doi.org/10.5194/bg-13-4291-2016, 2016.

Virkkala, A.-M., Aalto, J., Rogers, B. M., Tagesson, T., Treat, C. C., Natali, S. M., Watts, J. D., Potter, S., Lehtonen, A., Mauritz, M., Schuur, E. A. G., Kochendorfer, J., Zona, D., Oechel, W., Kobayashi, H., Humphreys, E., Goeckede, M., Iwata, H., Lafleur, P. M., Euskirchen, E. S., Bokhorst, S., Marushchak, M., Martikainen, P. J., Elberling, B., Voigt, C., Biasi, C., Sonnentag, O., Parmentier, F.-J. W., Ueyama, M., Celis, G., St.Louis, V. L., Emmerton, C. A., Peichl, M., Chi, J., Järveoja, J., Nilsson, M. B., Oberbauer, S. F., Torn, M. S., Park, S.-J., Dolman, H., Mammarella, I., Chae, N., Poyatos, R., López-Blanco, E., Christensen, T. R., Kwon, M. J., Sachs, T., Holl, D., and Luoto, M.: Statistical upscaling of ecosystem CO2 fluxes across the terrestrial tundra and boreal domain: Regional patterns and uncertainties, Global Change Biology, 27, 4040–4059, https://doi.org/10.1111/gcb.15659, 2021.

Zeng, J., Matsunaga, T., Tan, Z.-H., Saigusa, N., Shirai, T., Tang, Y., Peng, S., and Fukuda, Y.: Global terrestrial carbon fluxes of 1999–2019 estimated by upscaling eddy covariance data with a random forest, 7, https://doi.org/10.1038/s41597-020-00653-5, 2020.

Zhang, C., Brodylo, D., Sirianni, M. J., Li, T., Comas, X., Douglas, T. A., and Starr, G.: Mapping CO2 fluxes of cypress swamp and marshes in the Greater Everglades using eddy covariance measurements and Landsat data, Remote Sensing of Environment, 262, https://doi.org/10.1016/j.rse.2021.112523, 2021.

---

## Author Response (AR3)

**Comments to the author:**

The Referee recommends minor revisions and I agree with their assessment. Please address the remaining comments in a brief letter addressed to me and improve the manuscript accordingly and I would be happy to recommend its acceptance for publication in Biogeosciences.

Response to the editor: In the last round of minor revisions, we have addressed three minor comments from Referee #3. In light of your decision to address the remaining comments, in this round of minor revisions, we further refined our response to the third comment of referee #3 and improved the manuscript accordingly.

In addition, we have checked the manuscript for possible typos, etc.

**Response to remaining comments from Referee #3**

Comment 3. Please add an explanation for the reason why MLR, RF, SVM, and ANN are separately compared instead of all models and why PLSR with high R2 is removed in the Methodology section. This response is similar to the response for L198.

Last version Response: elaborated as 'Subsequently, the model accuracies corresponding to different levels of various features are compared in a cross-study fashion. In the evaluation of algorithms and time scales, we also implement comparisons within individual studies. For example, in the evaluation of the effects of the algorithms, we compare the accuracy of models using the same training data and keeping other features as constants in individual studies. In this intra-study comparison step, only algorithms with relatively large sample sizes in the cross-study comparisons were selected.'

Updated version response and revision: elaborated as 'Subsequently, the model accuracies corresponding to different levels of various features are compared in a cross-study fashion. In the evaluation of algorithms and time scales, we also implement comparisons within individual studies. For example, in the evaluation of the effects of the algorithms, we compare the accuracy of models using the same training data and keeping other features as constants in individual studies. In this intra-study comparison step, only algorithms with relatively large sample sizes in the cross-study comparisons were selected. In this study, algorithms with less than 10 available model records are not considered to have a sufficient sample size and we do not give further conclusive opinions on the accuracy of these algorithms due to their small samples (e.g., PLSR and BART with high R-squared but very few records as evidence). MLR, RF, SVM, and ANN were found to have large sample sizes (Fig. 5a), and thus their accuracies can be comparable. Based on this, in the intra-study comparison step, we only compare the accuracy differences between MLR, RF, SVM, and ANN in the context of using the same data and the same other model features (Fig. 5b).' **(Line 188)**